# A Citizen Science Approach to Identifying Indoor Environmental Barriers to Optimal Health for under 5s Experiencing Homelessness in Temporary Accommodation

**DOI:** 10.3390/ijerph19073976

**Published:** 2022-03-27

**Authors:** Diana Margot Rosenthal, Marcella Ucci, Michelle Heys, Antoinette Schoenthaler, Monica Lakhanpaul, Andrew Hayward, Celine Lewis

**Affiliations:** 1UCL Population, Policy and Practice Research and Teaching Department, Great Ormond Street Institute of Child Health, University College London, London WC1N 1EH, UK; m.heys@ucl.ac.uk (M.H.); m.lakhanpaul@ucl.ac.uk (M.L.); celine.lewis@ucl.ac.uk (C.L.); 2UCL Collaborative Centre for Inclusion Health, University College London, London WC1E 7HB, UK; a.hayward@ucl.ac.uk; 3UCL Institute for Environmental Design and Engineering, The Bartlett School of Environment, Energy and Resources, University College London, London WC1H 0NN, UK; m.ucci@ucl.ac.uk; 4Specialist Children and Young People’s Services, East London NHS Foundation Trust, London E15 4PT, UK; 5Center for Healthful Behavior Change, Institute for Excellence in Health Equity, NYU Langone Health, New York, NY 10016, USA; antoinette.schoenthaler@nyulangone.org; 6Community Paediatrics, Whittington Health NHS, London N19 5NF, UK; 7UCL Institute of Epidemiology and Health Care, University College London, London WC1E 7HB, UK; 8North Thames Genomic Laboratory Hub, Great Ormond Street Hospital, London WC1N 3BH, UK

**Keywords:** child homelessness, family homelessness, temporary accommodation, citizen science, inclusion health, indoor environmental quality, public health, inequalities, inequities

## Abstract

The first five years of life are critical for optimal growth, health, and cognitive development. Adverse childhood experiences, including experiencing homelessness, can be a risk factor for multiple health issues and developmental challenges. There is a dearth of data collected with and by families with children under age five living in temporary accommodation due to experiencing homelessness (U5TA) describing indoor environmental barriers that prevent U5TA from achieving and maintaining optimal health. The aim of this study was to address this current gap using a citizen science approach. Fifteen participants, who were mothers of U5TA living in a deprived area of London, and the lead researcher collected data in late 2019/early 2020 using: (I) a housing survey conducted via a mobile app; (II) house visits; and (III) collaborative meetings. Data were analyzed using thematic analysis. Key themes included: overcrowding/shared facilities, dampness/mold growth, poor/inadequate kitchen/toilet facilities, infestations/vermin, structural problems/disrepair, unsafe electrics, excessively cold temperatures, and unsafe surfaces that risk causing trips/falls, with all participants experiencing multiple concurrent indoor environmental barriers. The citizen science approach was successfully used to collect meaningful data demonstrating the need for child-centered housing policies meeting the needs of current and future generations of families living in TA.

## 1. Introduction

The first five years of life alone are paramount for optimal growth and development, including ~90% of brain development and achieving milestones (e.g., learning, speaking, walking) [1,2,3,4]. However, children under the age of five who live in temporary accommodation due to experiencing homelessness (U5TA) have a higher risk of developing chronic conditions and repeated cycles of homelessness and adverse childhood experiences (ACEs) throughout their lives [1,2,3,4,5,6,7]. Homelessness has increased at least three- to four-fold over recent years in high-income countries (HICs) such as the United Kingdom (U.K.) [8,9]. This problem is not confined to developed countries: estimates of children experiencing homelessness in developing countries have also highlighted the significance of the problem. However, these estimates, along with those collected in HICs, have been ambiguous with insufficient data collection, including real-time data and/or varying definitions of homelessness [9,10,11,12]. Therefore, child homelessness is a global problem that continues to grow [12]. On 18 June 2020, the United Nations (UN) Economic and Social Council adopted its first resolution on homelessness, which acknowledged there were multiple pathways leading “people of all ages from all walks of life” to an occurrence of homelessness in both developed and developing countries [12,13].

Studies on children experiencing homelessness have found that these children had higher odds of poor health, severe academic delay, behavior problems, and/or mental disorders with similar issues occurring in their parents/carers [9,14,15,16]. A recent scoping review article [9] found a series of interacting barriers preventing U5TA from accessing health services and achieving optimal health outcomes. However, the evidence found was sparse due to: (1) the lack of primary research studies on children under age five (under 5s) experiencing homelessness studied exclusively and (2) limited data collected by and with families living in temporary accommodation (TA) describing their experiences [9]. Citizen science is one approach to achieving inclusive studies and policymaking that can address these gaps.

“Citizen science”, first coined in the mid-1990s, has become an emerging area of research and practice where members of the public have a greater role within research and recognize that they play an invaluable role by providing insights not typically held by researchers reviving the “it takes a village” approach [17,18]. Citizen science is a form of community-based participatory research (CBPR) in the collaborative sense by involving all stakeholders in the research process of identifying a community need requiring transformational social change in order to improve health outcomes and mitigate health inequalities and inequities [19]. Although most studies have been ecological and conservational in nature (e.g., The “Arctic Hunters” Project) [20], citizen science has been framed as an important contribution to the democratization of medical care and healthcare [21]. Citizen science has been increasingly used to understand patient health experiences (e.g., PatientsLikeMe, a digital platform) [22], but few studies exist overall in the public health arena [21]. In the Netherlands, the iSpex project crowdsourced thousands of citizens to submit air quality measurements in order to assess the impact of atmospheric aerosols on health, climate, and air traffic through mapping [23]. Recently, the expansion of citizen science has been facilitated by digital technology specifically smartphones and the Internet. However, this approach fuels digital inequity, which mirror health inequities, therefore excluding clinically or socially vulnerable groups such as families living in TA from participating in citizen science approaches.

These aforementioned gaps were addressed by this study. The primary aim of this study was to describe the indoor environmental barriers to optimal health for U5TA. The secondary aim was to explore the suitability of a citizen science approach to address the primary aim.

## 2. Materials and Methods

### 2.1. Ethics

UCL Data Protection Office registration (no. Z6364106/2019/03/157) was obtained on 22 March 2019. UCL Research Ethics Committee (REC) approval ID number: 15097-001 was obtained on 30 May 2019. All benefits and potential risks associated with the study and working with a vulnerable group were identified and submitted as part of the UCL REC high-risk application process [24]. We also ensured that support for participants was available through facilitators, who were trained in trauma care and counseling, health visitors, and a nursery nurse if they felt that the study compromised their health or wellbeing.

### 2.2. Theoretical Frameworks

The study design, data collection materials, and analysis were guided by two theoretical frameworks: the social ecological model (SEM) [25] and health map for the local human habitat [26]. Together, these models provide a holistic view of how U5TA and their families are part of a global ecosystem and may be influenced by various determinants of optimal health, which include structural, systemic/political, environmental, socio-demographics, and more. The constructs within these models were used as the overarching framework for questions and observations conducted in the surveys and house visits, as well as to guide the overall themes in the analysis.

### 2.3. Study Design

This study adopted a citizen science approach whereby the research was co-created and participatory through the inclusion of citizens and their real-world problems along with the scientist as the co-designer and facilitator, which resulted in a shared, open, and reflexive research process [27]. This multi-methods study used methodological triangulation, including observations, a qualitative survey, photographs, and qualitative data collected from collaborative meetings (Figure 1). Methodological triangulation was chosen because it uses more than one kind of method to study a phenomenon. This helps to confirm findings and increase validity through a more comprehensive data set, thus providing a richer, more understanding of the phenomena being studied [28,29,30].

### 2.4. Study Setting

The London Borough of Newham (LBN), East London, U.K., was selected as the primary setting for this study because LBN has the highest number of homeless households in TA in London and England (49 per 1000 households) [31,32,33]. Temporary accommodation is included under the umbrella definition of homelessness, i.e., if a family is living in TA, they are experiencing homelessness [9,34], but not necessarily the other way around. Participants were conveniently sampled at a local charity, The Magpie Project, which provides support services for children under age five and their mothers experiencing homelessness who have access to limited resources to essentials such as nappies, baby supplies, clothes, travel expenses, and food bank referrals. At this venue, co-location of services was also frequently accessed by families living in TA both in and out of the borough. Socio-demographic data were to be provided by the charity to alleviate participant burden; however, due to events outside of our control, these data could not be accessed.

### 2.5. Study Overview and Co-Design

#### 2.5.1. Citizen Science Approach

This study was informed by European Citizen Science Association (ESCA)’s ten principles of citizen science to ensure best practices [35], which included working together with families with lived experience of TA in a community-based setting to generate new knowledge and understanding of critical issues that were currently impacting this vulnerable group. A key aspect of the citizen science approach was the preliminary meeting held in February 2019. Diana Margot Rosenthal (DMR) presented the aims of the project to a group of U5TA mothers (*n*~15 citizens), charity service users, and staff. This meeting and the involvement of potential participants at an early stage in the study development was to ensure that all stakeholders contributed to the study design. DMR discussed the ESCA principles in layperson terms, including the benefits of participation and potential outcomes of research dissemination, and, together with the charity, raised the issue of environmental health and access to health services. DMR pitched research question ideas to potential participants based on her discussions with cross-sector stakeholders and asked for feedback about which were the most important issues that needed public awareness. U5TA mothers discussed their challenges and anxieties of the TA environment and the impact that it could be having on their children’s health. DMR tailored the research question and objectives to reflect these concerns. Key co-design decisions also arose from this meeting, including the use of mobile smartphones and a mobile app for data collection, the use of the mobile app “WhatsApp” [36] to communicate during the study, and the feasibility of house visits to see TA housing conditions. The study had three parts: (I) Mobile App Housing Survey; (II) House Visits; (III) Collaborative Meetings.

#### 2.5.2. Part I: Mobile App Housing Survey

Participants collected data using a mobile app survey developed for this study. Questions were related to (1) participants’ TA environment (reported in this manuscript) and (2) aspects of the wider neighborhood (which will be reported elsewhere). The data elements pertaining to the housing environment captured per survey entry were: (1) photograph; (2) category of photograph location within housing; (3) descriptive text in response to the question “What are you showing in the photo?” and (4) first part of participants’ postcode. The free text descriptive captions were for participants to annotate the pictures with and reflect how the TA housing conditions shown in their image acted as barriers to their child’s health. Examples were given as prompts (Table 1), which were also collected in Part II (House Visits). Participants were able to submit multiple survey entries per category since only one photo and category could be recorded per entry.

Surveys were administered using the mobile app “ArcGIS Survey123” [37,38] (Appendix A). To mitigate the barrier of digital exclusion, each participant was loaned an identical smartphone with 15 GB of credit. A communal WhatsApp group was set up during the study to allow opportunities for participants to contact each other and provide any form of support, whether it was technological or social. Participants were assigned code names to maintain anonymity and reduce the risk of bias; these codes were used for survey submissions and in the WhatsApp group. Throughout the study, DMR closely monitored the group to ensure names and contact details stayed anonymized.

#### 2.5.3. Part II: House Visits

House visits were conducted by DMR to provide a more comprehensive account of the survey results. House visits included observational notes of the items recorded in the survey using an audio-recorder and digital photos. Participants guided DMR through the TA accommodation and pointed out issues that were of concern to them while DMR observed the environment more broadly. In addition, DMR also examined barriers that had been identified in the frameworks as well as a review of the literature [7] and the Housing Health and Safety Rating System (HHSRS) (Table 1) [39]. This included information about specific barriers, including safety risks, infestations, mold, infrastructural defects, poor ventilation, temperature control, and space (e.g., for a baby to crawl). U.K. local authorities use the HHSRS as a risk-based evaluation tool that assesses twenty-nine housing hazard categories ranked in order (1 = most serious and immediate risk to a person’s health and safety; 29 = least serious) and the effect that each may have on the health and safety of the occupants of a property [39,40]. During the analysis, DMR listed the prevalence of each hazard per house visit.

#### 2.5.4. Part III: Collaborative Meetings

Five collaborative meetings were held that occurred alongside data collection (Parts I–II) to gather participants’ feedback about the process, including any challenges they were encountering. The meeting was an informal, collaborative discussion lasting 45–60 min. Meetings were overseen primarily by DMR as well as charity staff, who were all trained facilitators when available. Participants were reimbursed for travel expenses and were given £10 vouchers for each collaborative meeting they attended. Three of the five meetings were audio-recorded and transcribed verbatim. The other two meetings were not recorded because they were shorter and were used as drop-in sessions to show participants how to use the app. We decided not to record these sessions as we wanted to keep them informal. Instead, DMR captured any important feedback through notetaking. In the collaborative meeting recordings, participants were coded as “Participant.CM1”, “Participant.CM2”, “Participant.CM3”, and so forth to maintain confidentiality.

#### 2.5.5. Pilot and Refinement of Study Tools

The mobile app housing survey was piloted over a two-week period with five participants. Following this pilot, in addition to feedback from two collaborative meetings, adjustments were made to the survey. This included the additional types of housing-related categories (e.g., ventilation, heating system) and word boxes where the participant could provide more detail around the submitted photo.

### 2.6. Recruitment

In September 2019, a small population of U5TA mothers was conveniently sampled from the Magpie charity with specific inclusion and exclusion criteria (Table 2). We decided only to recruit mothers (rather than fathers) for this study as they were the key demographic that the charity worked with, which included single parents and caregivers. The study was advertised during programming and workshops, with potential participants invited to leave their contact information on a sign-up sheet. During opening hours, DMR spoke with potential participants and provided them with written information about the study in English. Those interested in participating were asked to complete a consent form. Consent forms were not introduced until the participant had a chance to read or have the researcher read the information to them, understand the information, and ask any questions. At this time, participants were informed that they needed to return the phones at the end of the study.

### 2.7. Analysis

Data analysis was led by DMR with supervision from Dr. Celine Lewis (CL) and Dr. Marcella Ucci (MU). In order to assess the suitability of citizen science as an approach, i.e., “whether this approach produced reliable data that could be used for scientific purposes [42]”, we specifically looked at whether there was a concurrence between the surveys (conducted by the participants) and the house visits (conducted by the researcher). Comparing volunteer data with that collected by “professionals” has been identified as an important method of evaluating citizen science projects for data quality [42]. A further check was the collaborative meetings where findings were discussed among the group.

Each data set (survey, house visit, collaborative meetings) was analyzed separately in the first instance and then compared in the final stage during triangulation. We conducted thematic analysis using a six-stage framework [43]. Data familiarization was completed when all audio recordings were transcribed verbatim, photos saved, and survey responses downloaded from the database (Step 1). Initial a priori codes were generated using examples from Kingfisher’s Unfit Housing U.K. Research Report [44]. We also took into consideration how participants self-categorized the photos they had uploaded, which were sense-checked. Each survey and house visit photo was assigned a code (Step 2). These codes were collated into themes guided by the theoretical frameworks (Step 3). These were reviewed by CL and MU to ensure consistency within the themes (Step 4–5) [45] until no new themes emerged (data saturation). HHSRS hazards and their risks were then assigned to each theme so that the themes could be compared to an established framework of environmental indicators. Supportive quotations were pulled from the collaborative meetings when the participants and DMR discussed the study and various housing issues. Text taken from participants’ survey entries was recorded verbatim, and a word cloud of optional descriptions submitted by participants was created. Results were written up thematically and checked for agreement of the analysis with the supervisory team (Step 6) [43].

In order to triangulate the data, a two-step process was conducted: (1) whether the theme and HHSRS hazard were present; and (2) whether the same information and themes appeared across the three different collection methods and showed agreement [29,46]. A positive sign (+) indicated that the theme and hazard were both present, while a negative sign (−) indicated that the theme was present, but not the hazard. A double negative sign (− −) indicated that neither the theme nor the hazard was present. NA meant that the data were not applicable or not available to identify or assess the presence of a specific hazard.

## 3. Results

### 3.1. Mobile App Housing Survey

The pilot study occurred over two weeks (16 September 2019–30 September 2019). Five participants consented to participate, with three participants submitting 12 survey entries. The main study occurred over one month (16 October 2019–13 November 2019). A total of 15 participants consented to participate, and 11 completed the survey for a cumulative total of 48 entries. Overall, the 48 entries were reduced to 34 entries after the removal of duplicate/incomplete entries. The mean number of entries per participant was 4 (mode: 1, range: 0–12). Of the five pilot participants, four took part in the main study because the questions had been substantially revised; the fifth participant did not continue in the study because they were rehoused in TA outside of London and no longer met the inclusion criteria. One participant did not collect any data because they exceeded the 15 GB data credit limit early in the study period. Due to insufficient Wi-Fi access or time before the study concluded, other participants (*n*~6) had not finished submitting saved survey entries and/or ran out of the 15 GB airtime credit. The most frequently reported TA post codes were: E7, E13, IG3, IG5, N8, RM6, RM9, which indicated that participants also lived in TA outside LBN due to being relocated by local authorities in addition to the distances they traveled to access the charity’s free services.

### 3.2. House Visits

Four house visits were conducted between November–December 2019. House visits had to be rescheduled several times due to conflicting appointment times (e.g., GP, housing office, social services) and two rescheduled from January 2020 for February–March 2020, which were later canceled due to the onset of the COVID-19 pandemic. Over this period, at least two participants moved out of borough, and one participant dropped out on the first day due to being moved outside of London.

### 3.3. Collaborative Meetings

Five collaborative meetings were held on Wednesdays, 16 September 2019–25 November 2019. During Meeting 2, after the pilot had concluded, pilot participants were asked to comment on the app accessibility, functionality and usability, layout of the survey, and data collection measures. The participants recruited for the main study were also present and provided feedback based on print-out copies of the surveys. From this feedback, the participants and DMR determined what worked well and what did not, which led to further refinement. In Meeting 4, the main study was at the halfway point with the next round of participants. All participants (both pilot and non-pilot) discussed any issues or concerns and feedback regarding the survey and app changes with DMR. In the last meeting after the survey closed, participants were given printed copies of maps and images to see the data they had all been collecting; this was the raw, uncleaned data. Participants commented on aspects of the data they had collected and their experiences in the study. These comments fed into the analyses.

### 3.4. Thematic Analysis

Eight overarching themes were identified during the thematic analysis: (i) overcrowding and shared facilities, (ii) dampness/mold growth, (iii) poor/inadequate kitchen/toilet facilities, (iv) infestations/vermin, (v) structural problems/disrepair, (vi) unsafe electrical systems and appliances, (vii) excessively cold due to inadequate temperature regulation and (viii) unsafe surfaces that risk causing trips or falls. Each theme was matched to the corresponding HHSRS Categorical Hazard with a description of the health implications (Table 3). For each theme and HHSRS, results were triangulated to identify where there was agreement or disagreement across the three data collection methods (Table 3). The results showed significant agreement (+sign) of findings across all three data collection methods. The high level of agreement among data collected by the participants and researcher suggests that the data were of suitable quality and that the citizen science approach was suitable. This was further supported by the high level of agreement through collaborative meetings. During the analysis, some themes appeared to overlap because there was a causal relationship found between them [47]; for example, a structural problem or disrepair (e.g., broken windows) caused excessively cold temperatures. If there was overlap, each theme was analyzed on its own, and then we looked at what relationships existed. Themes needed to be analyzed separately because some did not have causal relationships.

#### 3.4.1. Overcrowding and Shared Facilities

Overcrowding and sharing facilities with different households was a consistent problem among most participants’ TA and found across all three data collection methods. In this study, overcrowding was defined as a lack of adequate space for the U5TA to play and explore and the OECD definition for “Housing Overcrowding” (e.g., the number of family members sharing a room) [48]. One cause of overcrowding reported by participants was that flats (including studio flats) were shared with housemates who were not family [49,50]. This was corroborated during the collaborative meetings, where participants discussed the difficulties of living with other people as highlighted by comments that they were “not good housemates” (speakers unknown) and that “Everyone has housemates. That’s why it’s temporary/shared accommodation. Everyone is going to have a housemate and share with at least one other family”. (Participant.CM7) These comments were confirmed during house visits, where overcrowding was clearly occurring. DMR observed a shared house of five families with two toilets and only one shower/tub; each family was assigned one of the five bedrooms, which created an issue of personal hygiene and no room for the children to move freely in the bedrooms. In addition, in two houses, there was no safe space for a baby to crawl, play or explore; therefore, the infants needed to stay in a buggy or highchair. This finding was supported by an image taken by a participant in the housing survey where they wrote: “That’s the space for my baby to crawl (Figure 2; Participant.Survey15)”.

Personal hygiene was another factor associated with overcrowding and sharing facilities with non-family members. Participant.CM7 described the everyday challenges of living in a shared house with so many families, especially with school-aged children. In this particular example, there were fifteen people living together in the shared house: four families with “10 school children and only two toilets…”. This mother described what it was like in the morning when everyone was trying to get access to the toilet: “There is rush to the toilet in the morning… it’s like a race against the other people because I’m living downstairs, there’s no toilet”. For this participant, the rush was also heightened because of the need to get their children ready and out of the house in time for school. Many participants also confirmed the poor ratio of toilets to occupants in several properties, with multiple school-aged children all needing to use the toilet around the same time. One participant, Participant.Survey17, expressed their upset through the housing survey about sharing a bathroom at a bed and breakfast (B & B) where the toilet was often left in an unclean state: “Just got in now and saw the mess that was made by the other housemates of the B & B”. Other issues associated with overcrowding included doing laundry. In surveys and collaborative meetings, participants in shared houses with one washing machine articulated the discomfort and challenges of having to wait five-plus days to do their laundry, especially with one or more U5TAs who use up their clean clothing and/or bibs very quickly during the week. In the surveys, 6 of the 11 participants, who submitted entries, said they experienced some sort of overcrowding and difficulties sharing facilities.

#### 3.4.2. Dampness/Mold Growth

Dampness and/or mold growth, a Category 1 Hazard, was a prominent theme and the most severe HHSRS category hazard found within TA properties across all data collection methods. Most survey participants (6 out of 11) wrote and photographed mold growth and dampness in their TA (Figure 3, Figure 4 and Figure 5, Surveys) and linked it to difficulties with their own or their children’s asthma during the collaborative meetings. Furthermore, participants reported mold growth in bathrooms and kitchens as well as bedrooms on their curtains (Figure 4, Participant.Survey10). Participant.Survey11 reported dampness that appeared to cover an entire wall, which was corroborated during the house visit (Figure 5). Mold and dampness were identified in three out of four house visits. Other participants reported mold growth and/or dampness in the collaborative meetings, including one participant who had not reported it in the survey because “I don’t want to show photos of my house because it’s not a good place... There is mould I wouldn’t want you to see that”. (Participant.CM6).

#### 3.4.3. Poor/Inadequate Kitchen/Toilet Facilities

The most common word that came up in the textbox submission during the housing surveys was “broken” (*n* = 10) (this doesn’t include phrases “not working”). After “broken”, the most frequently used words were “window”, “room”, “washing”, “buggy”, “damp”, “bugs”, and “stairs” (Figure 6). In one shared house, the participant reported a broken oven, broken refrigerator, broken washer/dryer, and broken kitchen cabinets (Figure 7 and Figure 8, Participant.Survey1). This was confirmed during the house visit, where the state of these major appliances was observed to be not in working order. Across all methods of data collection, washing machines were reported either broken or missing, which meant that the laundry had to be done in the shower or bathtub. Figure 9 (house visit) depicted a kitchen sink that was not working properly as well as water damage; the participant had to use the toilet sink for washing hands or dishes (see Section 3.4.5).

#### 3.4.4. Infestations/Vermin

Infestations and/or vermin were widely reported by participants as a common issue in TA. In the survey, Participant.Survey11 showed a cockroach infestation (Figure 10). This survey response was supported during the house visit, where cockroaches were seen throughout the TA. The tenants had clearly tried to address this problem by storing all their food in air-tight, sealed containers and by keeping the house tidy. Bed bugs were also a major concern. In Figure 11 from the survey, Participant.Survey13 discussed having an infestation of bed bugs in their TA: “There is a bugs in the room and bit me and my baby. I changed the room to another room but the problem still all the house has bugs. [sic]”. Similarly, in the collaborative meetings, one participant mentioned the frequency with which mattresses were changed in TA, highlighting that this was a significant issue since multiple occupiers were using the same mattress each time a new TA resident moved.

#### 3.4.5. Structural Problems and/or Disrepair

Most TA had structural defects or damage, which could be potentially unsafe and unhealthy indoor living environments. In Figure 12 (Participant.Survey1), the participant reported the floorboard coming up next to the one shower/bath being shared by five families. During the house visit, Figure 13a,b (house visit) were taken, and DMR clarified the extent of this issue as a tripping hazard increasing the risk for serious physical injuries, but also for water to seep underneath the floor and produce further mold exposure, which could be seen on the ceiling ventilation fan. This house was in disrepair from every corner and had each one of the HHSRS environmental hazards listed in Table 3. In the same TA, the extent of a large reported “Cracked wall” (Participant.Survey1) that extended from the floor to ceiling on the top floor (Figure 14, house visit) was observed, which appeared to be water damage coming from the roof, and made the participant feel distressed about it being a structural issue and becoming worse with time and/or feeling neglected. In a different participant house visit, a mother and son (~2 years old) lived in a studio located in a hidden alleyway next to a garage in LBN; the toilet facility had cracks in the walls and missing floor tiles (Figure 15, house visit). The participant also commented that they had difficulty with washing their child in that shower with the small opening.

In almost every survey, participants reported a broken window that was being kept closed with cellophane/duct tape: “I have a broken glass window in the toilet as well the aluminum [sic], the window stays open I cannot shut it (Figure 16, Participant.Survey10)”. As a result of the broken windows, TAs were excessively cold, and some were even made worse by a broken boiler, Participant.Survey10; see Section 3.4.7).

#### 3.4.6. Unsafe Electrical Systems and Appliances

Unsafe electrical systems and appliances were a prominent finding across the data collection methods. Multiple TA sites documented exposed electrical outlets (Figure 17, house visit) or wires coming out of the walls. Overall, many TAs were not suitable for under 5s and were not childproofed. For example, during one house visit, a participant reported that when they took a shower, they needed to put their son in the highchair to protect him from the numerous environmental hazards, including easy access to electrics, e.g., the stove knobs at a standing level for the child (Figure 18, house visit). The kitchen was also wide open in this studio space and not childproofed, therefore was a constant risk that a child could turn the stove on which could lead to the child burning themselves or causing a fire.

#### 3.4.7. Excessively Cold Due to Inadequate Temperature Regulation

Many participants reported broken windows and/or boilers causing excessively cold temperatures, a Category 2 Hazard, which could be detrimental to child health. Participant.Survey10, a mother of two U5TA, had been trying to get the local authority to repair their broken boiler for more than a month (Figure 19). They also described the window above their son’s bed: “Window is broken and the rooms are very cold because of this” (Figure 20, Participant.Survey10). A recurring theme across all forms of data collection was the need to use cellophane tape to patch up windows (Figure 21, house visit) and false doors, which were possibly former windows or fire escapes (Figure 22, house visit) to keep the cold air from coming in. This was further corroborated during the house visit (Figure 21) and the indoor temperature was comparable to the outdoor temperature 5/6 °C that day [51]. The participant informed DMR that their son had sickle cell disease and feared that this environment only made him more ill. A total of 3 out of 11 participants who completed the surveys reported these issues; when they did not report them, they discussed them in the collaborative meetings, so more than 50% of participants were experiencing these problems.

#### 3.4.8. Unsafe Surfaces That Risk Causing Trips or Falls

Unsafe surfaces with changes in level greater than 300 mm risked causing trips or falls. There was agreement across the data collection methods that TA housing had dangerous staircases leading to the property, within the property, or both (Figure 23, Figure 24, Figure 25, Figure 26, Figure 27 and Figure 28. A total of 7 out of 11 participants in the surveys reported these issues. In Figure 23, Participant.Survey15 described: “That is the staircase leading to my apartment. It is very strenuous to go on this staircase into my apartment especially carrying my baby. One day, I nearly fell on the stairs with my baby”. Participant.Survey16 reported that the “Iron was slippery” in reference to the stairs leading to the property. During Participant.Survey16’s house visit, DMR also photographed the stairs (Figure 24, house visit) and climbed the stairs; which were very steep, damp, and appeared to be fire escapes; this could be particularly dangerous for a mother carrying their toddler plus buggy in the rain (Figure 25, house visit).

Inside TA properties, the stairs were very steep, and no safety provisions were in place (e.g., child gate), nor accessible in the case that a child had a disability (Figure 26 and Figure 27, housing surveys; Figure 28, house visit). Different participants reported the difficulties and inconvenience of these stairs with the buggy, especially since there was nowhere on the ground floor to park it: “The stairs is not convenient for buggy, whenever we are going out we have take buggy down first, especially when we go for shopping, is really difficult [sic] (Figure 27, Participant.Survey10);” and “The stairs are inside the property and it is very dangerous for kids [sic] and I have to take the buggy upstairs every time (Figure 26, Participant.Survey12)”.

## 4. Discussion

The aim of this study was to describe the indoor environmental barriers to optimal health for children under the age of five experiencing homelessness by virtue of living in temporary accommodation (U5TA). This is the first citizen science study that we are aware of to explore indoor environmental barriers with mothers of U5TA in the U.K. and worldwide. Our findings revealed that the most commonly reported barriers were dampness/mold growth, overcrowding and shared facilities, and unsafe stairs that could cause trips or falls. Barriers occurred in clusters and compounded each other (e.g., broken window, broken boiler, structural disrepair), giving rise to a variety of problems and sometimes making each subsequent one worse (e.g., excess cold, hazards from window not being able to be shut, broken glass, mold/dampness). Although some of these hazards were known to exist in unfit housing before, none had been studied directly using a citizen science approach and with this marginalized group.

### 4.1. Benefits and Ethical Challenges of Citizen Science

The second aim of this study was to determine the suitability of a citizen science approach to address the primary aim through data reliability. We found numerous benefits to using a citizen science approach in a community-based setting. Participants benefited by gaining experience of scientific methods, developing social capital in their own community of U5TA mothers, and demonstrating a sense of empowerment [52]. We were able to involve participants in the study design, data collection, and analysis, ensuring the findings were grounded in their experiences. Using this approach, we were able to co-design a new method of data collection, namely the mobile housing survey app, which coincided with the house visits. Tasks were designed to be simple to promote high accuracy, minimize bias, account for the demanding schedules of the participants, and easy to navigate regardless of technical skill. The collaborative meetings that were conducted during data collection enabled us to take participants’ views and feedback into account, as well as provide an opportunity for participants to discuss the emerging findings, thus facilitating an iterative approach to the design and conduct of the study. This improved the openness and reliability of the research and increased the participation and engagement of citizens in the scientific research. In addition, there was a better understanding of the citizens’ marginalization as well as their inclusion in informing policy and practice, a group normally excluded from both [21,52,53,54,55,56]. While our ambition had been to involve participants in further policy recommendations and results dissemination, unfortunately, this was not possible due to the pandemic.

Although there are many benefits to citizen science, it also raises a few ethical concerns and challenges. One concern is data quality and integrity, meaning the data collected by participants may not meet scientific standards due to a lack of participant training in scientific data management or research [52,57,58]. To address this concern and ensure reliable data that could be used for scientific purposes [42], we checked for concurrence between the surveys (conducted by the participants) and the house visits (conducted by the researcher) with the collaborative meetings where findings were discussed among the research team used as an additional check. Another potential concern is around data integrity and intellectual property [42]. Throughout the study, the aims and objectives, as well as rights and ownership of the data, were communicated clearly and openly with participants and other stakeholders through participant information sheets and continuous feedback and interaction with the researcher. Results were made publicly available and shared with the LBN local authority, with the aim that this would be a steppingstone toward transformational change. While working with a vulnerable group, the loaning of phones and distribution of vouchers raises potential ethical concerns around coercion. In order to address this concern, we were very clear from the outset that the phones were on loan only, and that the vouchers were an honorarium payment in acknowledgment of their contribution and time. The barrier of digital poverty was mitigated by loaning all participants identical smartphones with the same amount of data credit ensuring equal and equitable participation in the study. This practice also provided that equipment used for measurements were standardized and calibrated across participants. For future studies, researchers should also consider these ethical challenges and plan accordingly when conducting citizen science studies with vulnerable groups.

### 4.2. Health Implications

Many of the indoor barriers identified in this study have been shown to be significant hazards that have a detrimental impact on health. For example, dampness and mold, a Category 1 Hazard, increases the risk for allergies, asthma, exposure to toxins from mold and fungal infections, and potential threats to mental health and social wellbeing [39,59,60,61,62]. Asthmatic attacks, allergic reactions, and infections (from any over 40 types of pathogens) can also be triggered by bed bug infestations, another health hazard found in this study, whereby multiple occupiers were found to use the same mattress for various lengths of time [39,63]. U5TA were often found to be left in a buggy or pram as a safety provision because of lack of space and other hazards (e.g., mold, pests/vermin), but this can critically hamper development and meeting health milestones, even posing a significant and powerful risk for progression to a neurodevelopmental disorder or syndrome if not addressed [64,65]. Broken windows, structural problems/disrepair, unsafe electrics, staircases with a change in level greater than 300 mm, and the lack of safety provision (e.g., baby/child safety gates) were all found in this study and increase the risks of injury, collision or entrapment in addition to being markers of social deprivation and health inequality [39,66]. Finally, we found that indoor temperatures in TA were excessively cold, which is a Category 2 Hazard, and increased risk of respiratory conditions of flu, pneumonia, and bronchitis for the entire household and cardiovascular conditions of heart attacks and strokes for the adults [39].

### 4.3. Contribution to the Literature

Many of these findings have been identified elsewhere, however, using different methodologies. A recent report by the Children’s Commissioner in England found TA were overcrowded, excessively cold, “frequently not fit for children”, and the report concluded that “poor quality TA presents serious risks to children” [9,67]. In 2001, an audit into homeless families conducted by health visitors in the U.K. found that indoor environmental barriers such as shared toilets and kitchen facilities spread infection and encouraged an unhealthy diet [9,68]. Similar audits conducted more recently in the U.K., in 2018 and 2020, reported the health implications for families living in TA (poor nutrition, higher hospital admission, and poor mental health); however, these audits did not address any of direct causes from the built environment [69,70]. A contribution of this study is that it addresses this gap by providing visual evidence accompanied by the lived experience of the barriers to health that occur in the indoor housing environment, conducted in collaboration with experts by experience (i.e., U5TA and their mothers).

### 4.4. Policy Implications

This study has highlighted the importance of policy alongside greater monitoring of and accountability for the safety and regulations of TA to ensure that these environments promote optimal growth and development for U5TA to thrive in [45]. Ideally, local authorities should carry out a full inspection and a hazard assessment under the HHSRS before deciding if accommodation offered to an applicant is suitable [71,72]. In particular, they must verify that any accommodation is free of Category 1 hazards (Dampness/mold growth) and is fit for human habitation [73,74]. In this study, the lack of basic safety provisions for any of the hazards (e.g., unsafe electrics, broken windows, steep stairs, gas cooktops) showed these TA were not suitable and detrimental to the health and development of the children. As a transient population, it was difficult to imagine that the participants were solely responsible for safety provisions when they could be moved to a new TA with different housing layouts at short notice, could not afford such preventive measures, and/or did not know how long they would be living in that TA, i.e., three months or three years. Suitable housing conditions and availability must come to the forefront on the national agenda since the COVID-19 pandemic has likely exacerbated many of these risks and hazards by increasing the rates of homelessness.

### 4.5. Strengths and Limitations

A strength of this study is the citizen science approach with methodological triangulation, which facilitated a fuller, more complete picture of the topic. Furthermore, triangulating the findings from the three methods increased the validity of the findings [75] and provided a bi-directional, mutual exchange of ideas, learning, and resources as well as collective understanding [76], which is the ultimate goal of community engagement [77]. Citizen science projects have been prone to similar biases such as traditional biomedical research and speak “about” rather than “with” vulnerable groups, but we ensured that this citizen science project reduced social exclusion [21,76]. Rather than a top-down approach of having an auditor come in and tick off a box on a list, all these issues were seen through the lens of those living there and experiencing these problems, which were barriers to optimal health. The ECSA’s 10 principles were used and informed the design of the study to ensure we adhered to best practices [35].

A limitation of the study was the sample size and use of convenience sampling, which possibly increased the likelihood of selection bias; however, many participants lived and traveled in and out of the borough to access the charity as well as other resources, and due to their transient circumstances, this was the best approach to sampling and recruitment. Due to limited funding of the study, translations of all materials and interpreters could not be provided, which limited the diversity of the TA population that could be represented. The inability to obtain the socio-demographic data was also a limitation in this regard, which would have provided important context to this study as potential indicators of health inequalities and inequities [78,79]. However, while the sampling approach and lack of demographic information mean that the study lacks objective information on the generalisability of the findings, many aspects are still of interest beyond the study sample itself. This innovative work can inform and facilitate evidence-based policy, further supporting the 2020 Marmot Review [6] that housing has an impact on health and is a driver of health inequalities and inequities [80,81].

On a usability level, participants reported frustration with the inability to take more than one photo per survey and the need to submit multiple entries on the same indoor space (i.e., stairs, bathroom, kitchen) along with other mobile app features, which were all technological limitations of the app itself, combined with their other daily competing priorities. In the surveys, the number of entries was lower than expected, suggesting a nonresponse bias. Fewer house visits were conducted than anticipated due to the pandemic, but also participants later changed their minds and/or did not want to show their TA because they were embarrassed, did not want to get in trouble/feared altercation with their roommates/housemates, or both, which were issues that came to light during the collaborative meetings. Because of the pandemic, contact with the participants was lost, so they were not able to participate in further analyses and dissemination of the results. Unfortunately, the pandemic had an unprecedented impact on vulnerable groups, which could not be controlled for.

## 5. Conclusions

The early years are a short yet vital period to ensure the next generation have the best start in life; however, as highlighted by this study, U5TA faces numerous indoor environmental barriers that can have significant short- and long-term impacts on their health and wellbeing. Through a collaborative approach with U5TA families, this study showed how housing conditions affected their daily life and ability to care for their children. This study demonstrated how the co-presence of multiple established environmental hazards and poor living conditions as experienced by the citizens created “the perfect storm”, which adversely affects families and their children within a vulnerable group who are already experiencing several challenges. Future studies should continue to use citizen science as well as co-production, which are evidently suitable and important approaches to address health inequalities influenced by social determinants and the built environment by working together with experts with lived experience and ensuring their inclusion. Based on difficulties reported by participants about the survey, future citizen science studies in the public health field should consider all factors (accessibility, functionality, and usability) when developing the protocol and choosing the best mobile app for data collection, building on the latest evidence from the literature. Finally, future studies should also consider the use of citizen science approaches to data collection, which occur over a longer time period and with participants taking an active role in data collection as well as dissemination of results, including policy recommendations.

## Figures and Tables

**Figure 1 ijerph-19-03976-f001:**
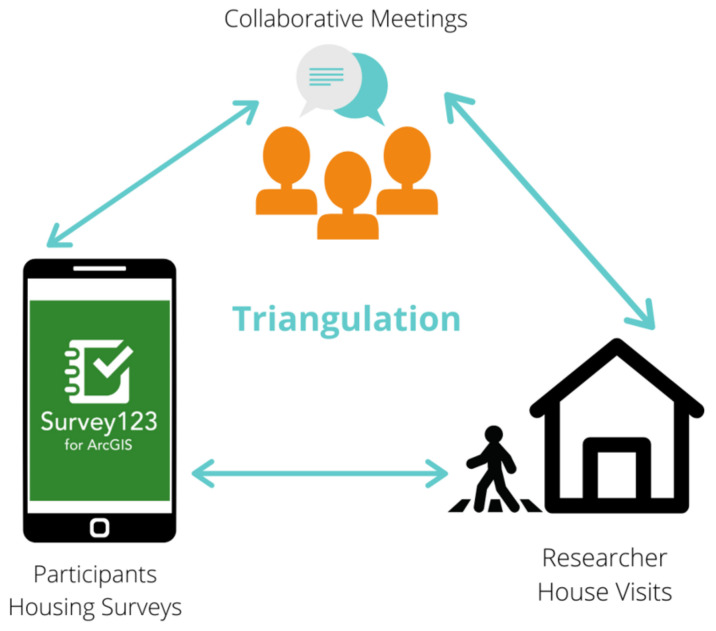
Methodological triangulation in study design.

**Figure 2 ijerph-19-03976-f002:**
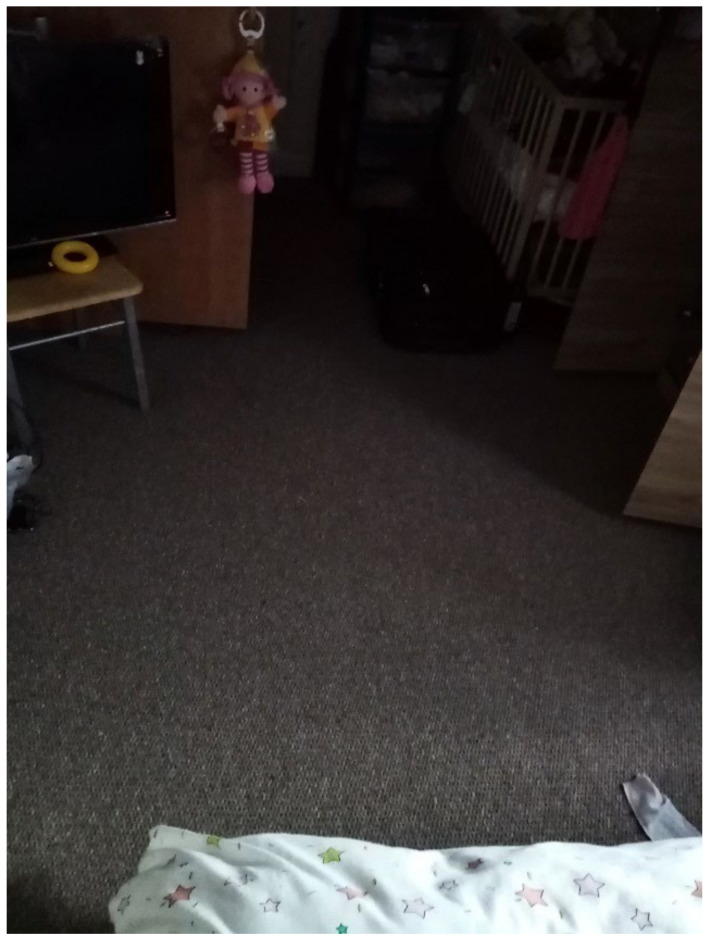
“That’s the space for my baby to crawl.”—Participant.Survey15.

**Figure 3 ijerph-19-03976-f003:**
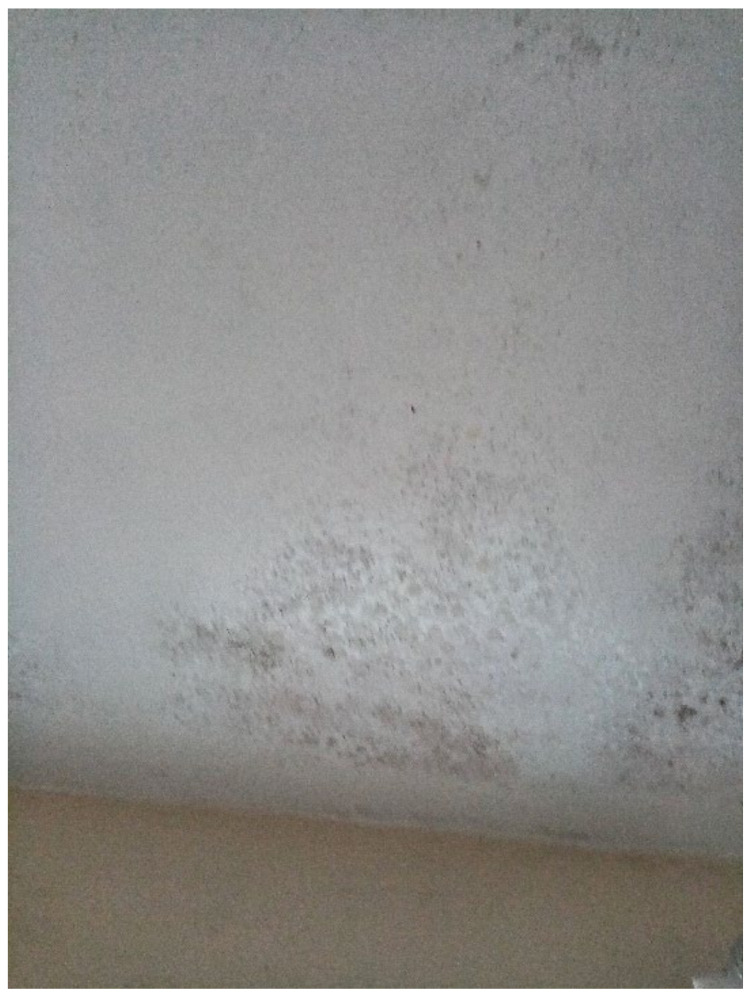
“Mould everywhere.”—Participant.Survey10.

**Figure 4 ijerph-19-03976-f004:**
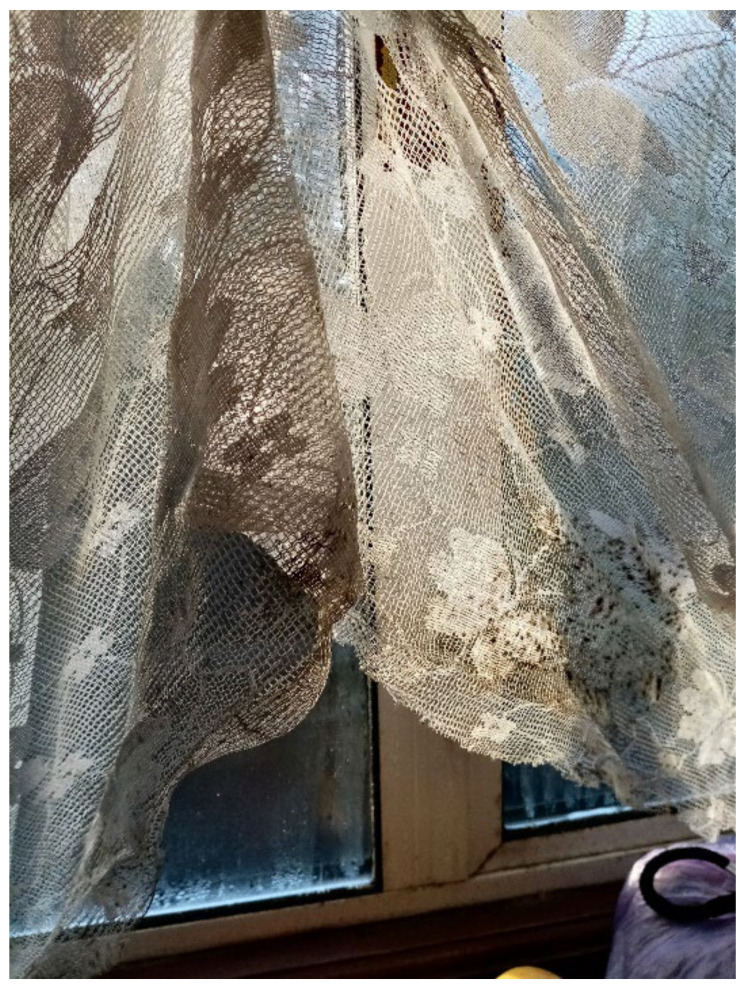
“Mould in the curtains.”—Participant.Survey10.

**Figure 5 ijerph-19-03976-f005:**
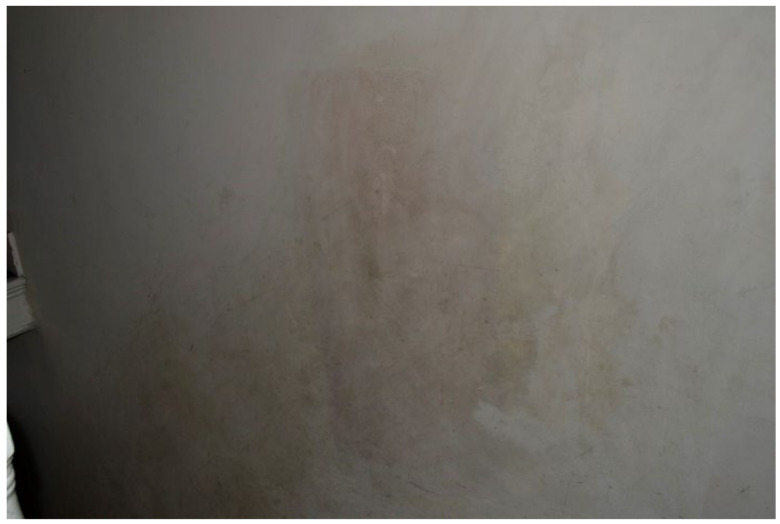
House visit where Participant.Survey11 reported “Damp a wall” [sic].

**Figure 6 ijerph-19-03976-f006:**
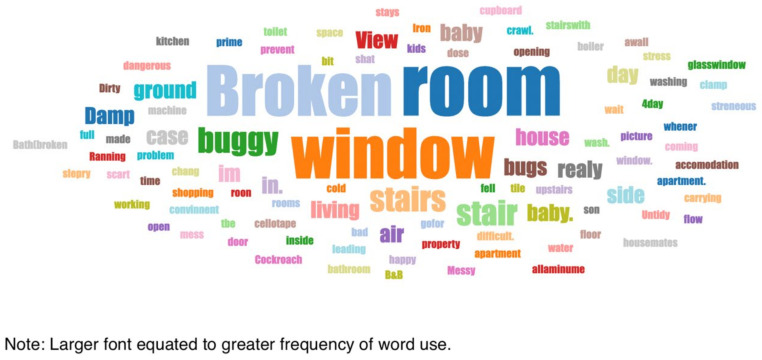
Word cloud of optional accommodation descriptions.

**Figure 7 ijerph-19-03976-f007:**
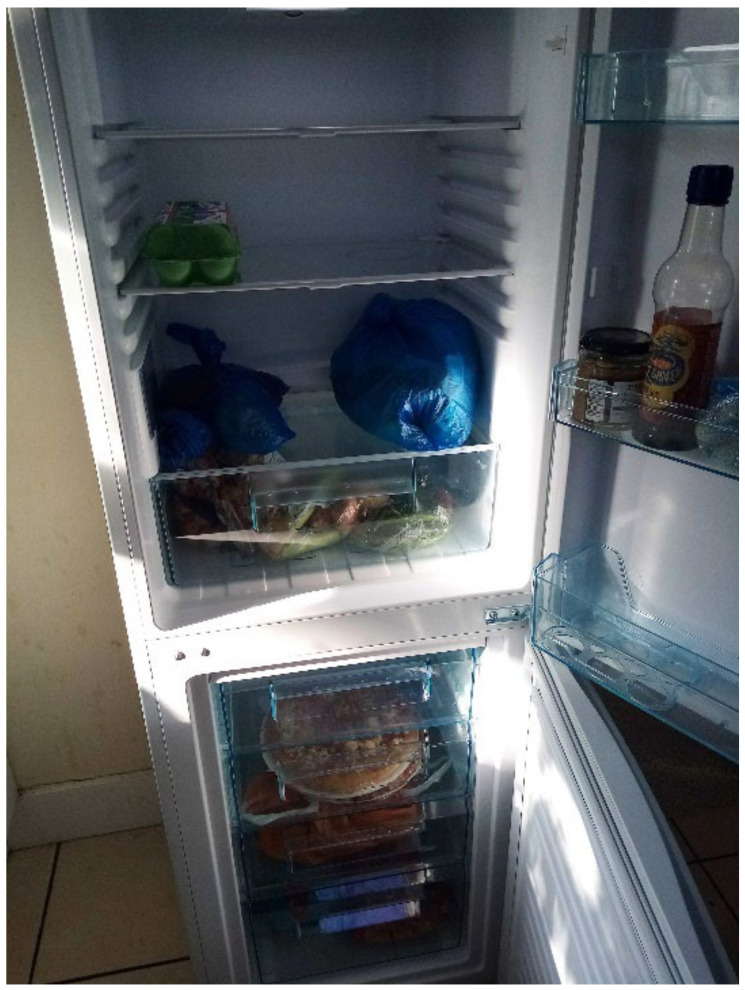
“Broken fridge”—Participant.Survey1.

**Figure 8 ijerph-19-03976-f008:**
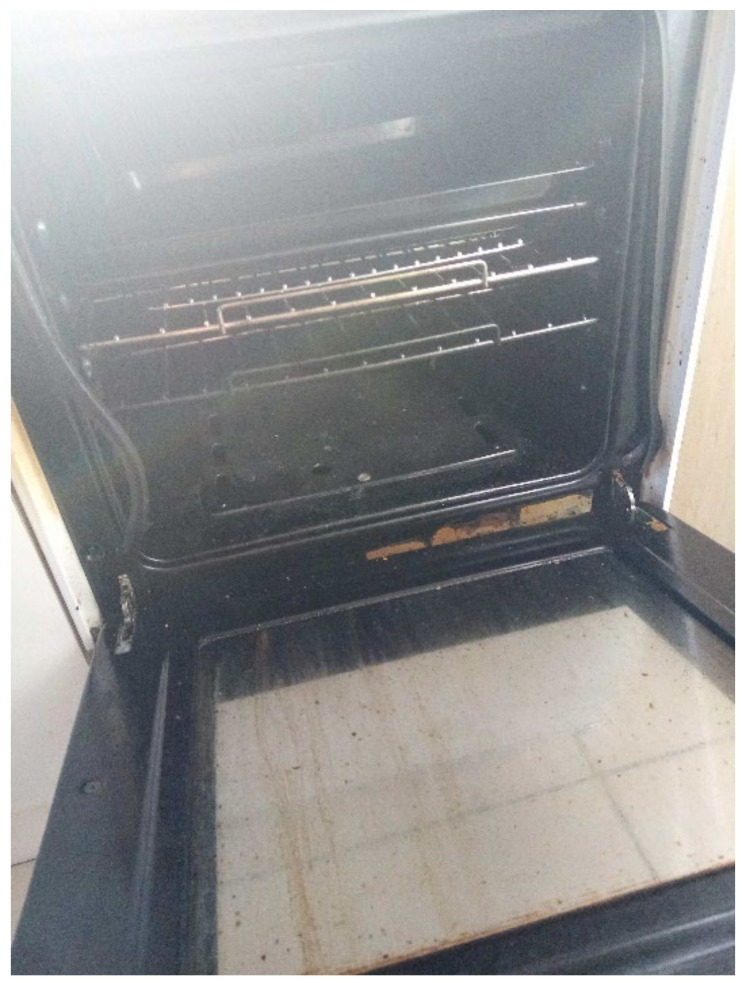
“Broken oven”—Participant.Survey1.

**Figure 9 ijerph-19-03976-f009:**
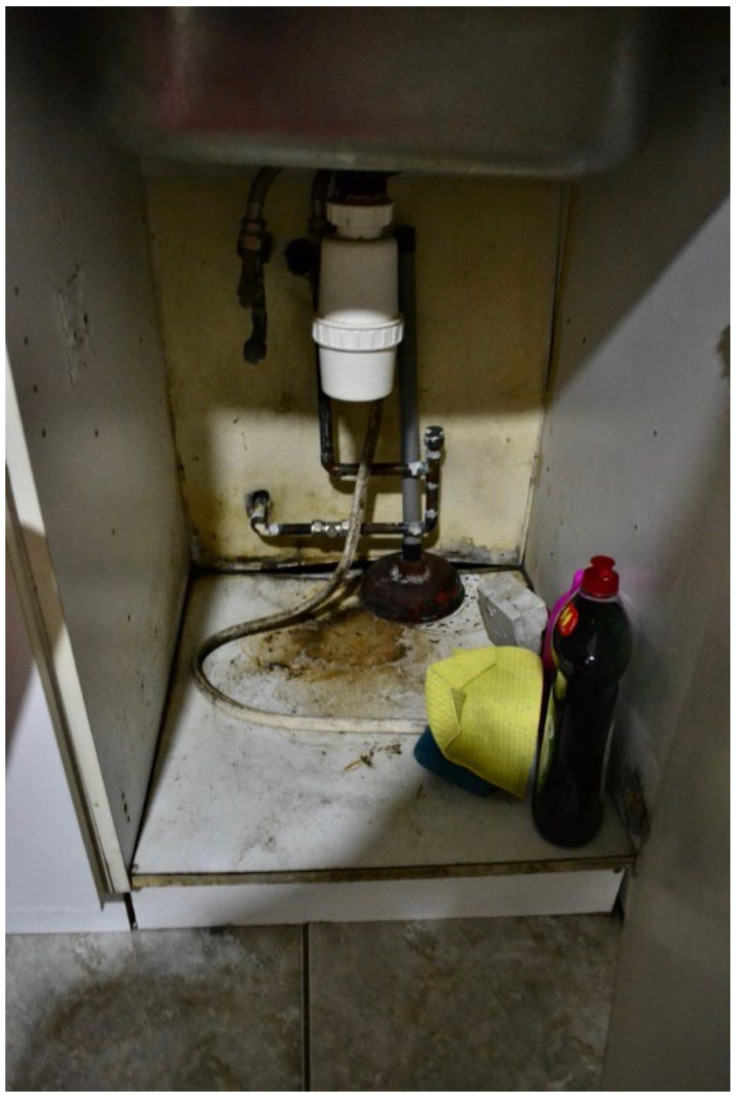
House visit.

**Figure 10 ijerph-19-03976-f010:**
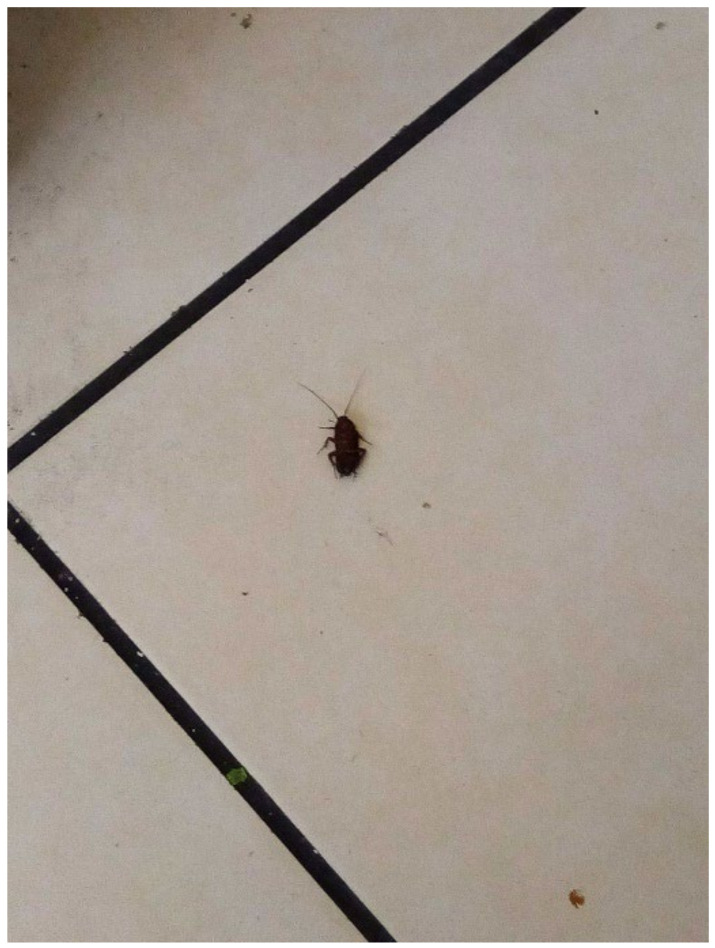
“Cockroach”—Participant.Survey11.

**Figure 11 ijerph-19-03976-f011:**
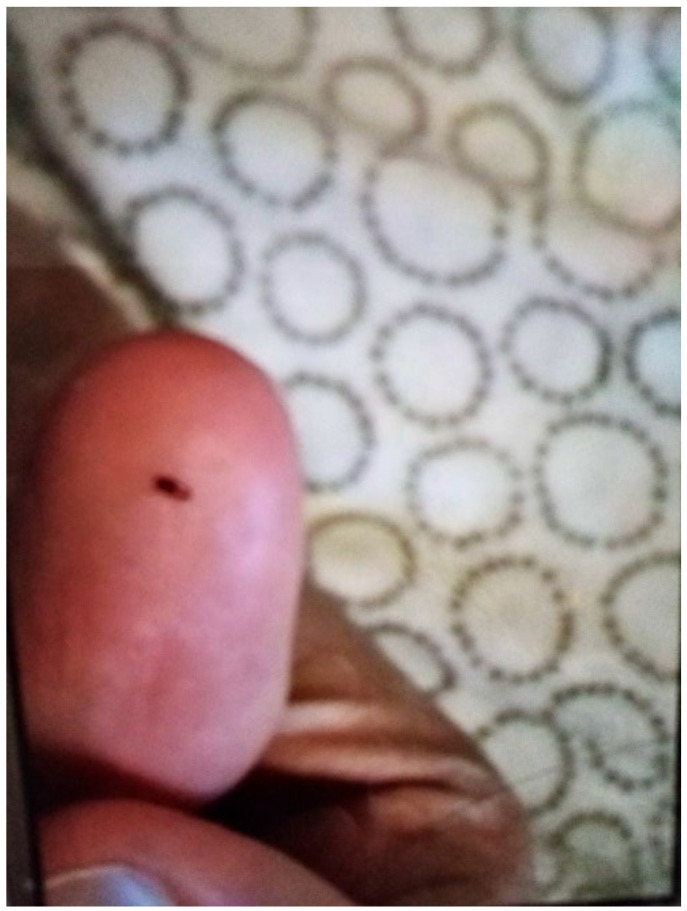
Participant.Survey13.

**Figure 12 ijerph-19-03976-f012:**
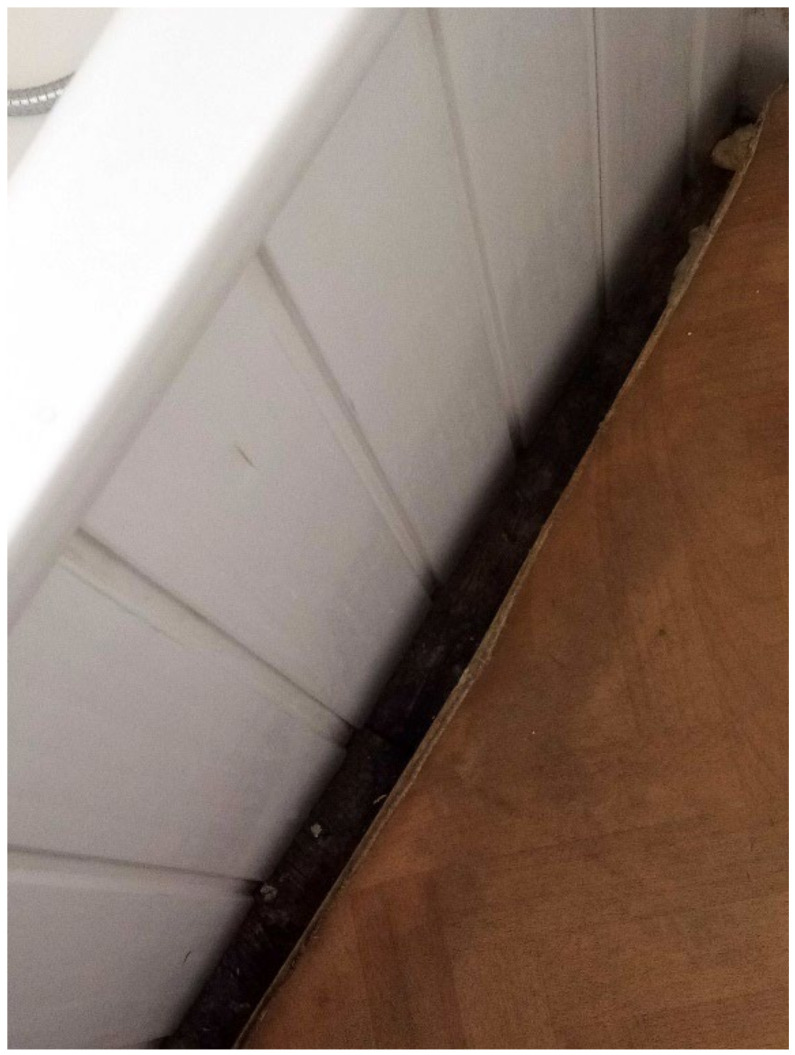
Participant.Survey1.

**Figure 13 ijerph-19-03976-f013:**
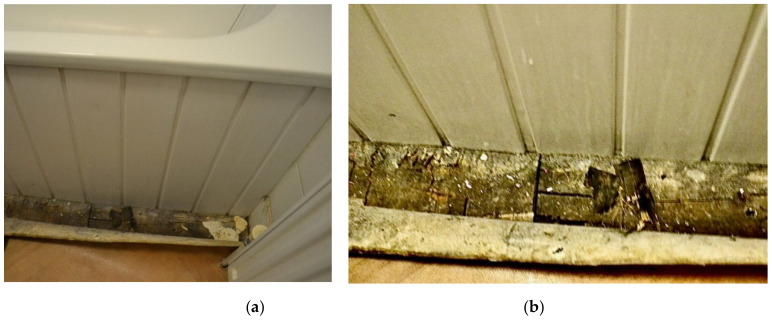
(**a**) House visit. (**b**) House visit.

**Figure 14 ijerph-19-03976-f014:**
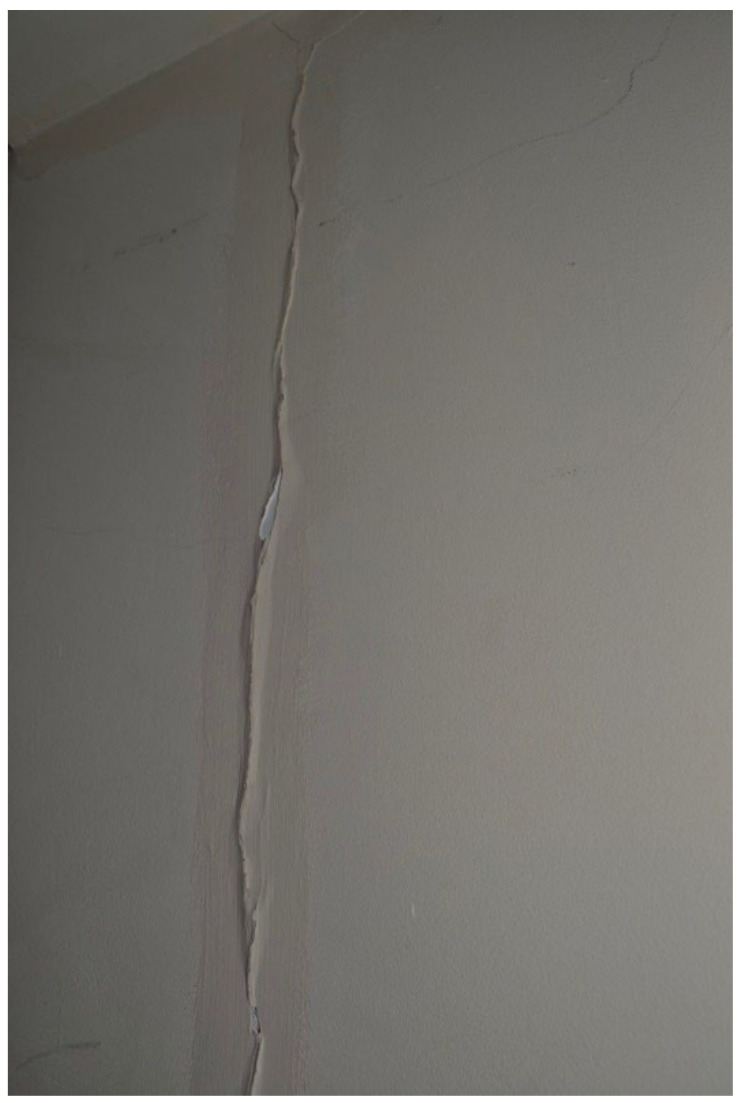
House visit.

**Figure 15 ijerph-19-03976-f015:**
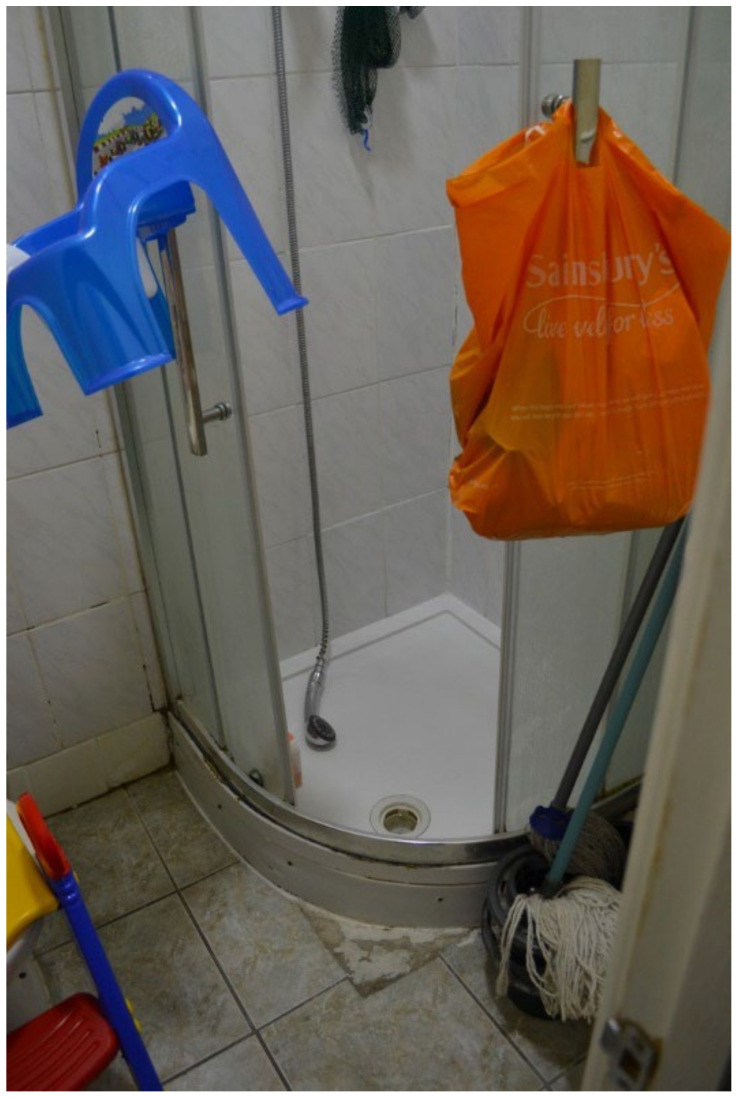
House visit.

**Figure 16 ijerph-19-03976-f016:**
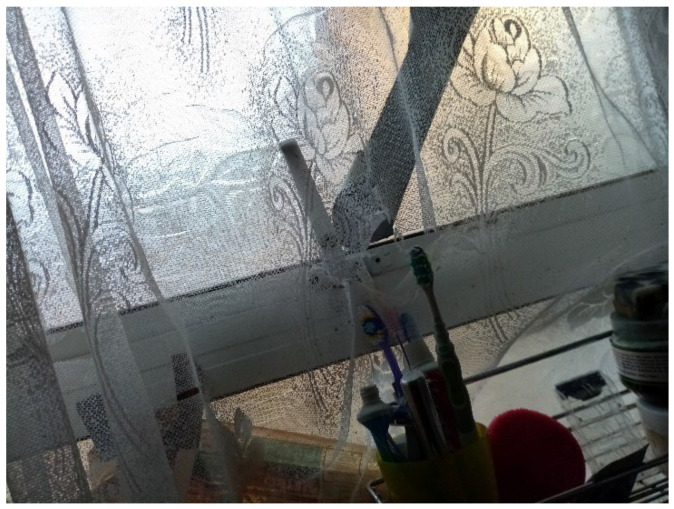
“Broken windows and humidity”—Participant.Survey10.

**Figure 17 ijerph-19-03976-f017:**
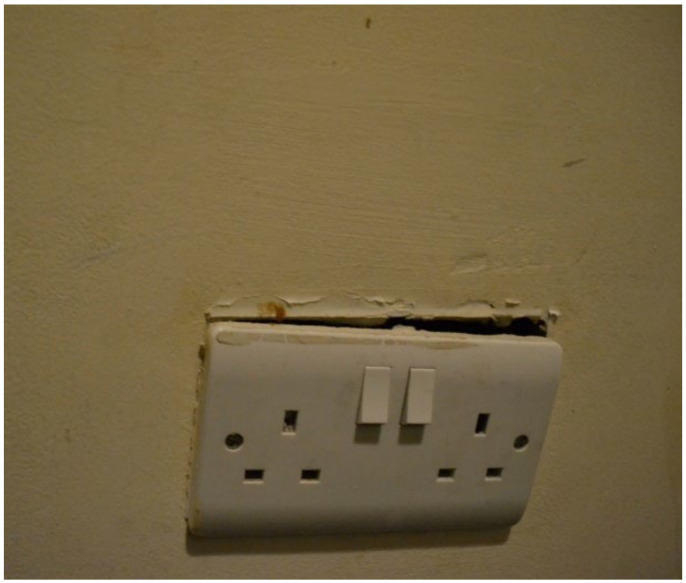
House visit.

**Figure 18 ijerph-19-03976-f018:**
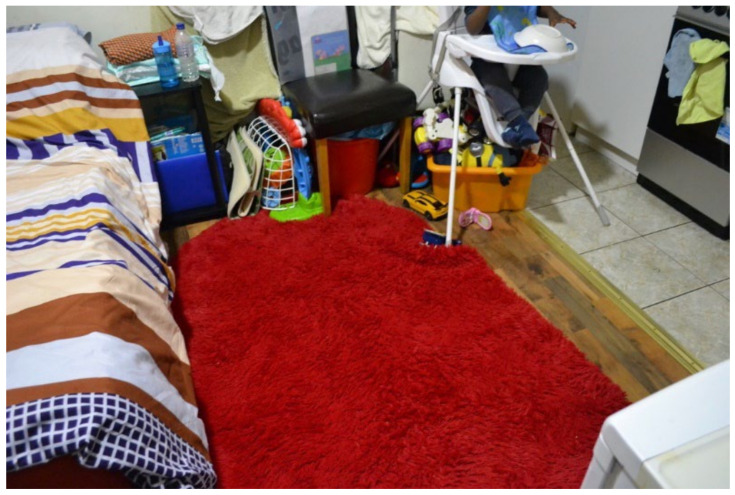
House visit.

**Figure 19 ijerph-19-03976-f019:**
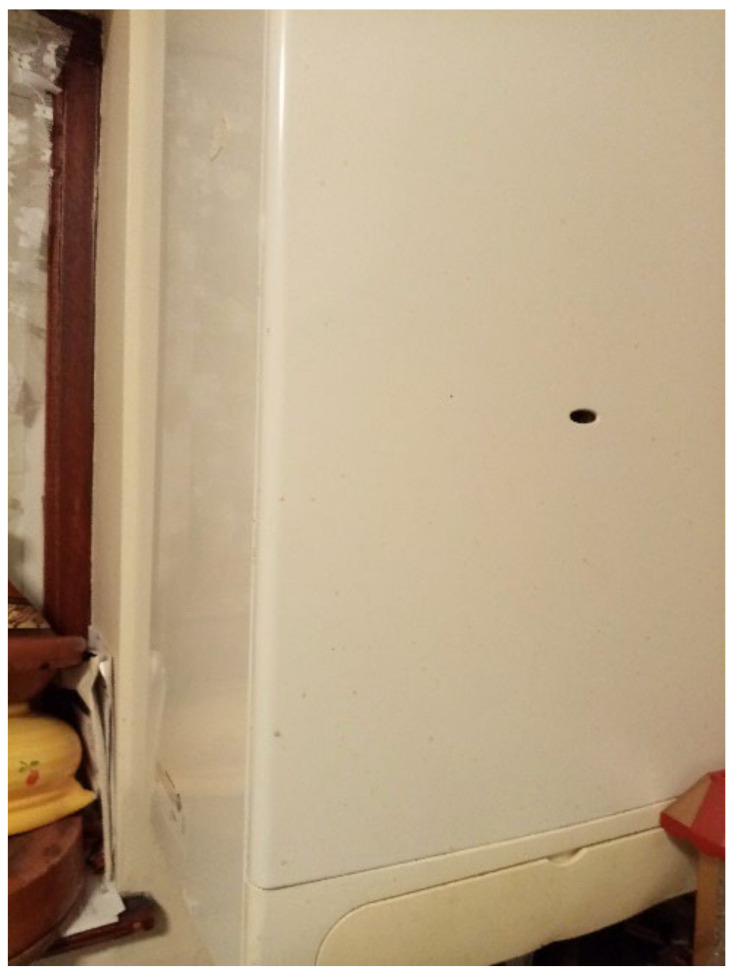
“Broken boiler. The boiler is not working”. Participant.Survey10.

**Figure 20 ijerph-19-03976-f020:**
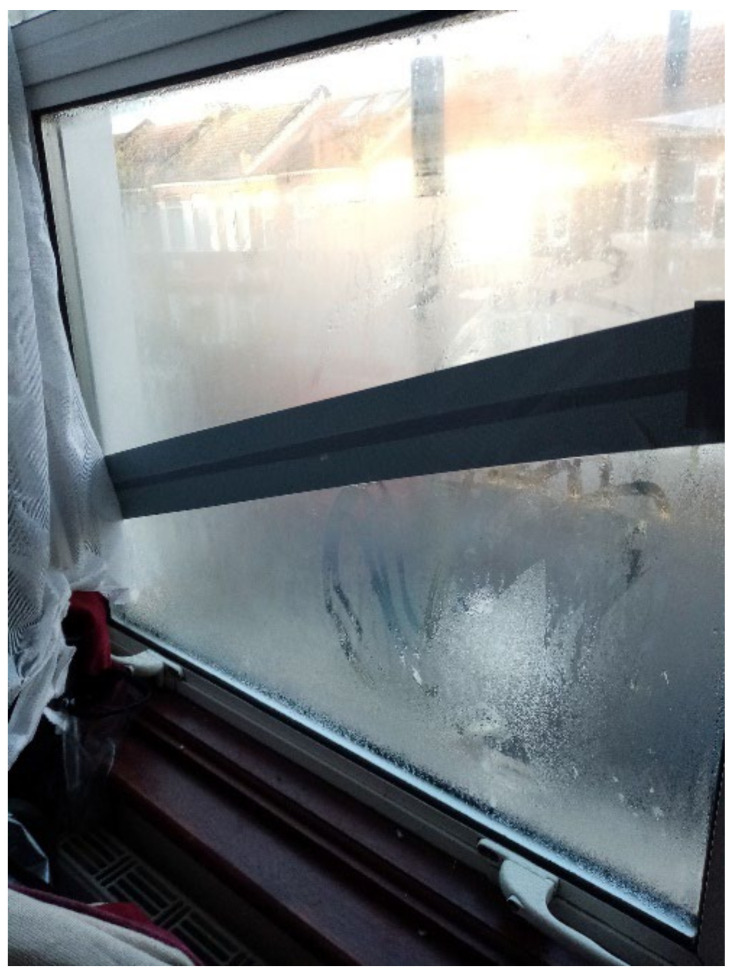
Participant.Survey10.

**Figure 21 ijerph-19-03976-f021:**
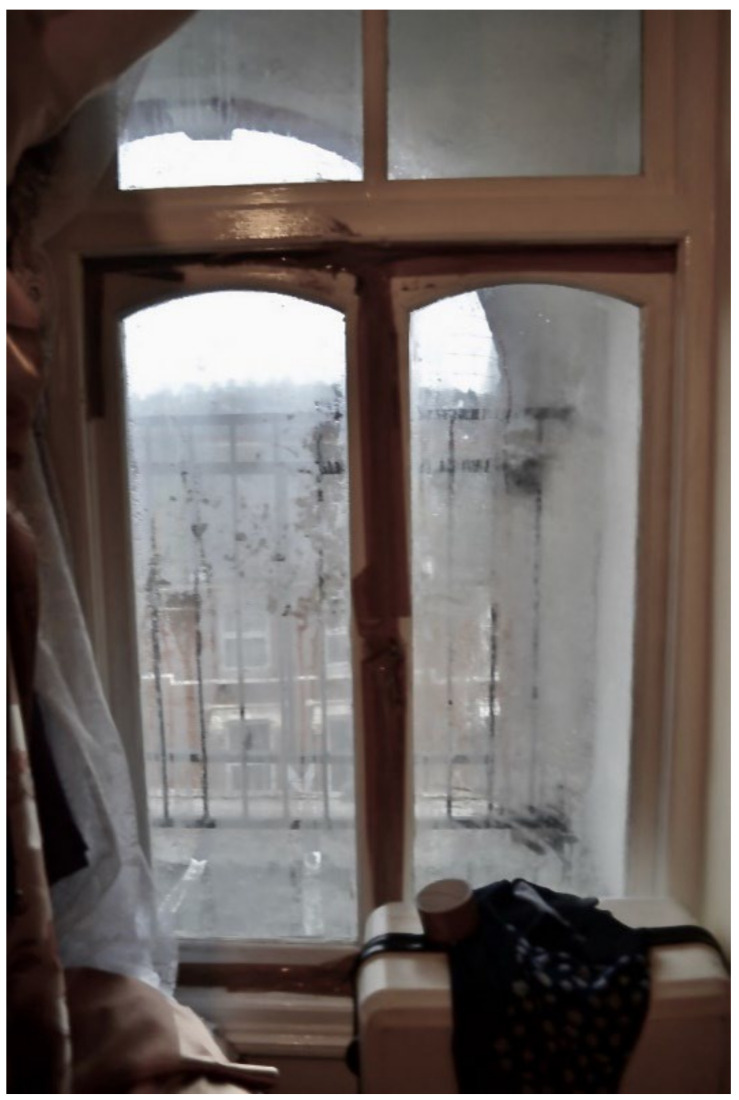
House visit.

**Figure 22 ijerph-19-03976-f022:**
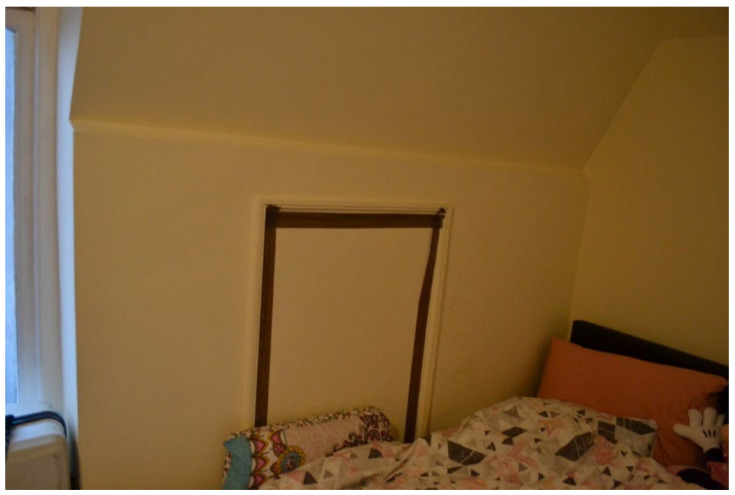
House visit.

**Figure 23 ijerph-19-03976-f023:**
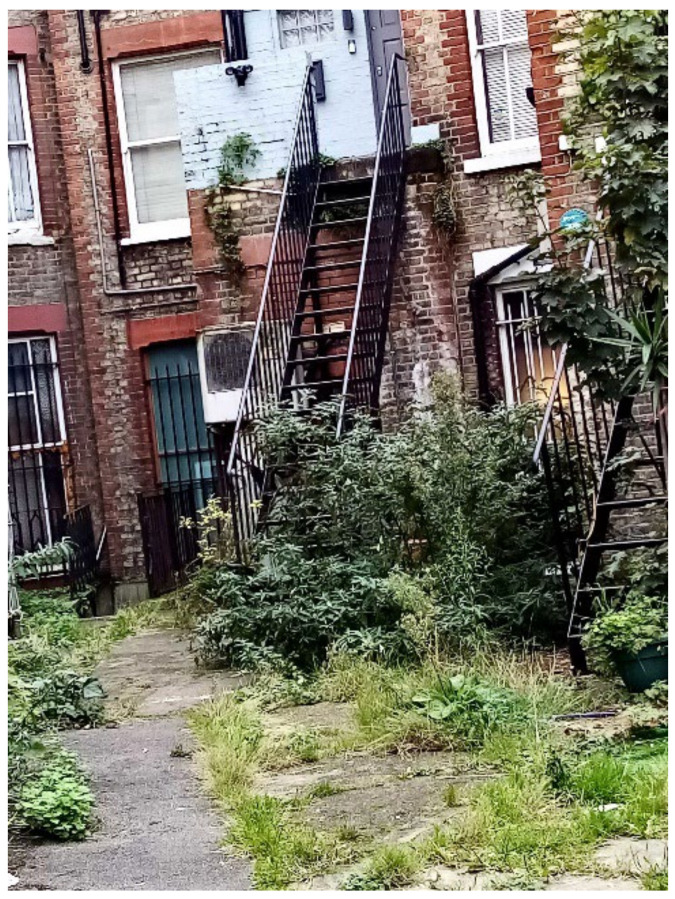
“The staircase leading into my apartment.”—Participant.Survey15.

**Figure 24 ijerph-19-03976-f024:**
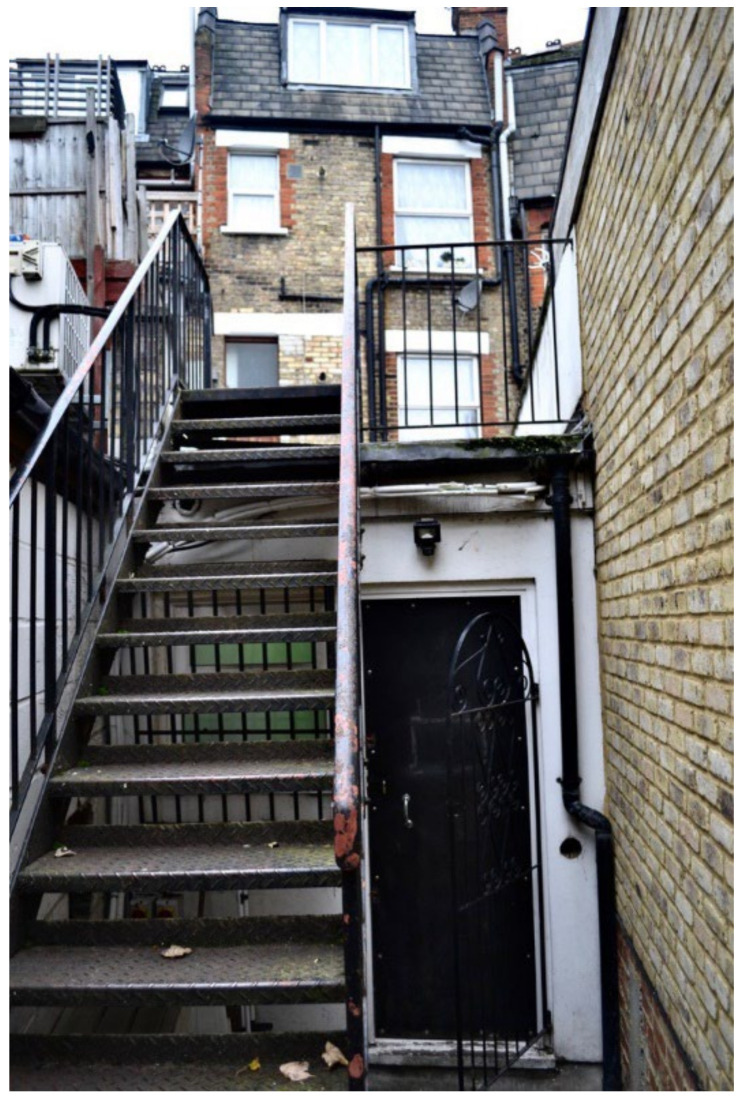
House visit.

**Figure 25 ijerph-19-03976-f025:**
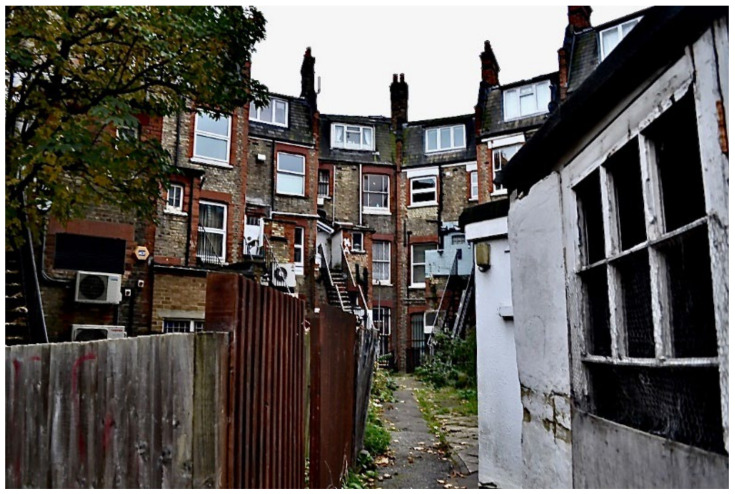
House visit.

**Figure 26 ijerph-19-03976-f026:**
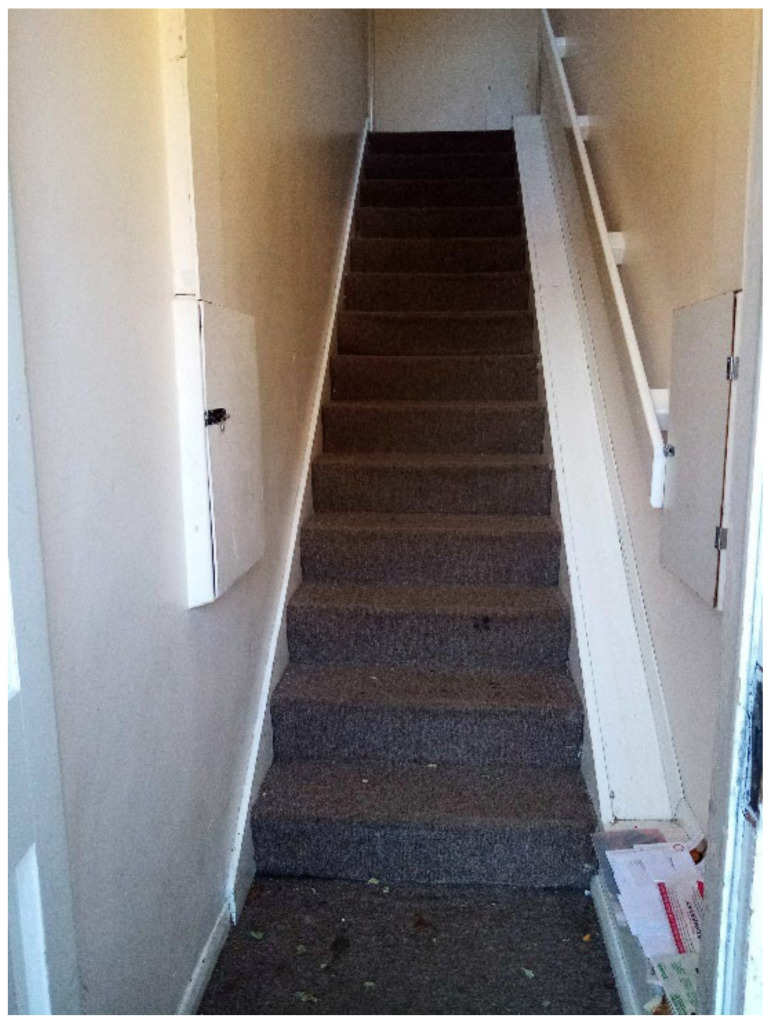
Participant.Survey12.

**Figure 27 ijerph-19-03976-f027:**
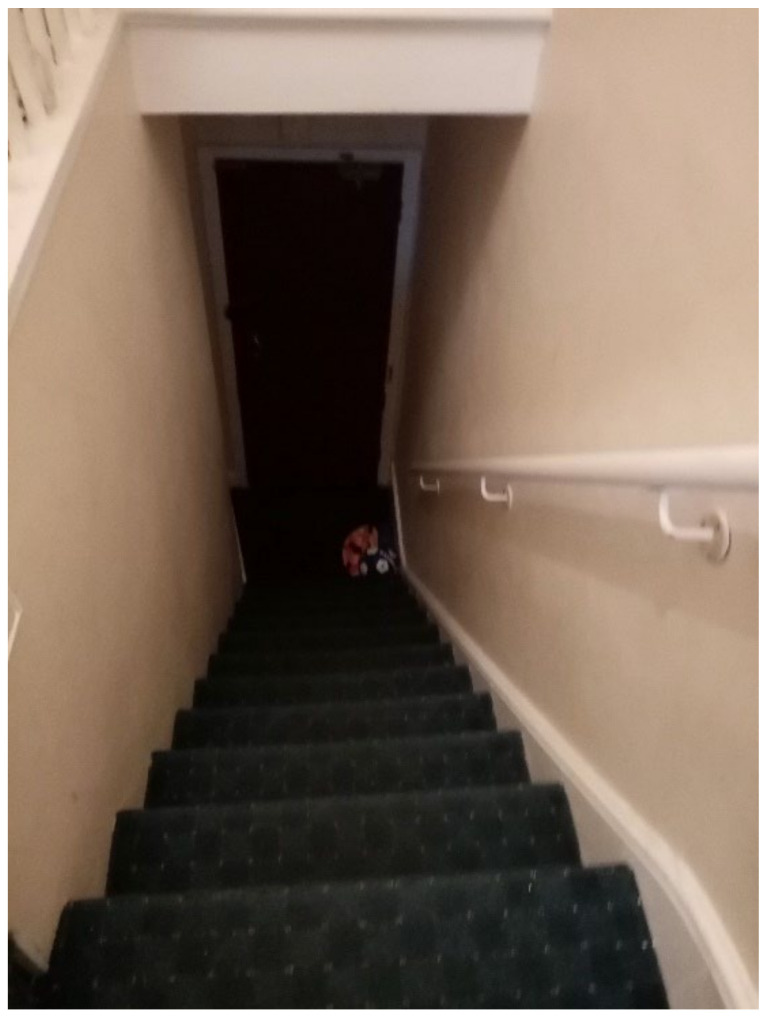
Participant.Survey10.

**Figure 28 ijerph-19-03976-f028:**
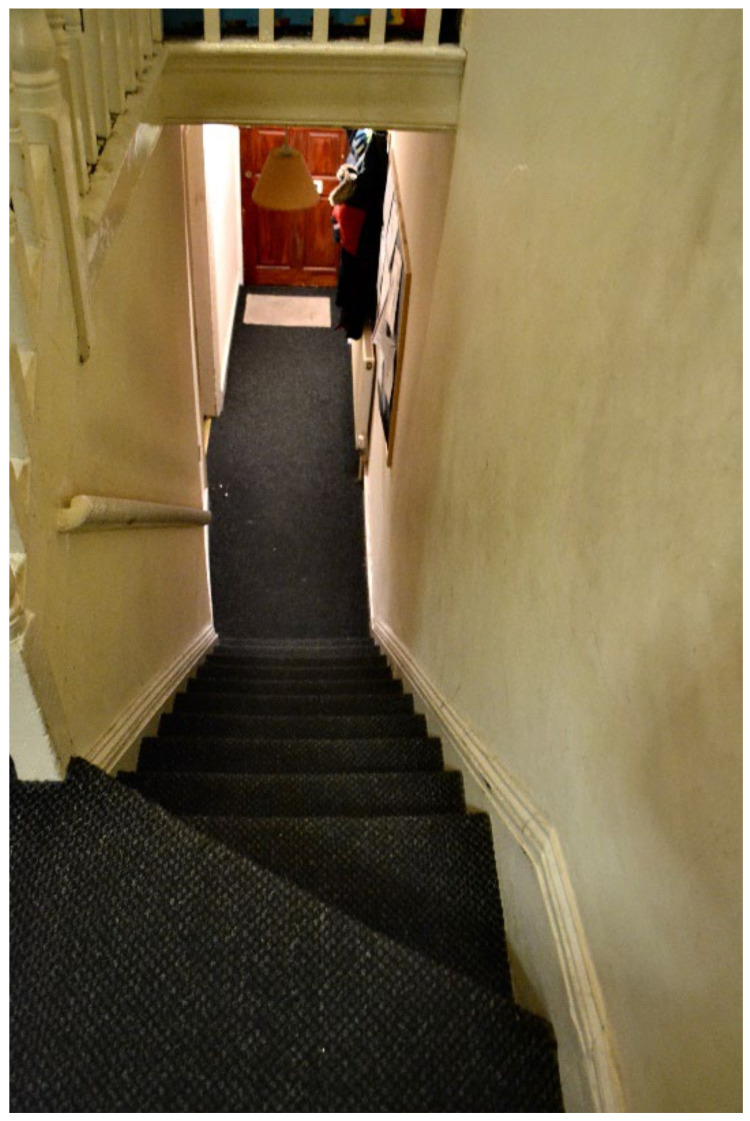
House visit.

**Table 1 ijerph-19-03976-t001:** Observation data collected in house visits.

Description	Observation Data Collected
Built Environment—Housing Community-Level Barriers and Facilitators Photographs Audio-recorded observation notes	Type of temporary accommodation and housing conditions Ease of access to and within the property Safety risks Infestations/vermin Dampness/mold Structural defects Noise Temperature control, poor ventilation Space (e.g., for a baby to crawl) The condition of the restrooms and the number of people sharing it The size/condition of shared rooms The location of the kitchen; private or shared Cleanliness

**Table 2 ijerph-19-03976-t002:** Participant criteria for pilot and main study.

Inclusion Criteria
• Be a service-user of the charity
• Be in temporary or insecure accommodation * at the time of the study
• Be staying in London in a Newham or non-Newham postcode
• Pregnant and/or mother of children aged <5 years
• Ability to communicate in English
• Be willing to send/receive text messages
• Be willing to have a house visit by the researcher
• Be willing to collect data on a mobile app
• Be >16 years of age
**Exclusion Criteria**
• Refuse or are unable to provide informed consent
• Father of children aged <5 years
• Have significant psychiatric comorbidity, cognitive impairment, which may impair ability to provide informed consent (as documented in The Magpie Project’s client records)
• Plan to discontinue support and services at the charity within the study period

* Note: The European Federation of National Organizations working with the Homeless (FEANTSA) definition, the European Typology of Homelessness and Housing Exclusion (ETHOS) [41].

**Table 3 ijerph-19-03976-t003:** Multi-methods triangulation of housing environment findings.

Thematic Category	HHSRS Hazard Category and Description	Health Effects (Taken from HHSRS)	House Visits	Participant Surveys	Collaborative Meetings
I. Overcrowding and Shared Facilities	**11. Crowding and space** Hazards associated with lack of space for living, sleeping, and normal household or family life	-Psychological distress and mental disorders, increased risk of hygiene issues, accidents, and personal space and privacy compromised. -Increased risk of infection (e.g., COVID-19)	+	+	+
	**13. Lighting** Threats to physical and mental health associated with inadequate natural or artificial light, including the psychological effects associated with the view from the property through glazing	Depression and psychological effects due to lack of natural light. Eye strain from glare and inadequate light	+	+	NA
	**14. Noise** Threats to physical and mental health due to exposure to noise within the property or within its curtilage	Psychological and physiological changes resulting from lack of sleep, poor concentration, headaches, and anxiety	+	NA	+
	**17. Personal hygiene, sanitation, and drainage** Threats of infections and threat to mental health associated with personal hygiene, including personal and clothes washing facilities, sanitation, and drainage	Stomach and intestinal disease, skin infections, and depression	+	+	+
II. Dampness/Mold Growth	**1. Damp and mold growth** Health threats due to dust mites, mold or fungal including mental and social wellbeing health threats associated with damp, humid, and moldy conditions	Allergies, asthma, effects of toxins from mold and fungal infections	+	+	+
III. Poor/Inadequate Kitchen/toilet Facilities	**16. Food safety** Threats of infection from poor provision and facilities to store, prepare, and cook food	Stomach and intestinal disease, diarrhea, vomiting, stomach upset, and dehydration	+	+	+
	**19. Falls associated with baths** Falls associated with a bath, shower, or similar facility	Physical injuries: cuts, lacerations, swellings, and bruising	+	+	NA
IV. Infestations/Vermin	**15. Domestic hygiene, pests, and refuse** Health hazards due to poor design, layout, and construction, making it hard to keep clean and hygienic, attracting pests, and inadequate and unhygienic provision for storing household waste	Stomach and intestinal disease, infection, asthma, allergies, disease from rats, and physical hazards	+	+	+
V. Structural Problems/Disrepair	**26. Collision and entrapment** Risks of physical injuries from trapping body parts in architectural features such as trapping fingers in doors and windows and colliding with objects such as windows, doors, and low ceilings	Physical injuries such as cuts and bruising to the body	+	+	+
	**29. Structural collapse and falling elements **** The threat of the dwelling collapsing, or part of the fabric being displaced or falling due to inadequate fixing or disrepair or as a result of adverse weather conditions	Physical injuries	+	+	+
VI. Unsafe Electrics	**23. Electrical hazards** Hazards from electric shock and electricity burns	Electric shock and burns	+	+	+
	**24. Fire** Threats to health from exposure to uncontrolled fire and associated smoke. It includes injuries from clothing catching fire, a common injuring when trying to put a fire out	Burns, being overcome by smoke, or death	+	+	+
	**25. Flames, hot surfaces, and materials** Burns or injuries caused by contact with a hot flame or fire, hot objects, and non-water-based liquids. Scalds caused by contact with hot liquids and vapors	Burns, scalds, permanent scarring, and death	+	+	NA
VII. Excessively Cold Due to Inadequate Temperature Regulation	**2. Excess cold **** Threats to health from cold indoor temperatures. A healthy indoor temperature is 18 to 21 °C	-Respiratory conditions: flu, pneumonia, and bronchitis -Cardiovascular conditions: heart attacks and strokes	+	+	+
VIII. Unsafe Surfaces That Risk Causing Trips or Falls	**20. Falls on the level surfaces** Falls on any level surface such as floor, yards, and paths, including falls associated with trip steps, thresholds, or ramps where the change in level is less than 300 mm	Physical injuries: bruising, fractures, head, brain, and spinal injuries	+	+	NA
	**21. Falls associated with stairs and steps** Falls associated with stairs and ramps where the change in level is greater than 300 mm. It includes internal stairs or ramps within a property, external steps or ramps associated with the property, access to the property and to shared facilities or means of escape from fire and falls over stairs, ramp or step guarding	Physical injuries: bruising, fractures, head, brain, and spinal injuries	+	+	+
	**22. Falls between levels** Falls from one level to another, inside or outside a dwelling where the difference is more than 300 mm. Including falls from balconies, landings, or out of windows	Physical injuries	+	+	+
	**28. Ergonomics** Threats of physical strain associated with functional space and other features at the dwelling	Strain and sprain injuries	+	+	+

NA stood for either not applicable or not available. For example, there was no data applicable in participant surveys for the thematic category overcrowding under Hazard 14 (Noise) because there was not a way to assess noise from the survey photos, nor did the participants report it in the textbox. A positive sign (+) indicated that the theme and hazard were present, while a negative sign (-) indicated that the theme was present, but not the hazard. ** Note: This is based on the participants and DMR reporting and observations of large cracks in the walls and foundation in addition to significant moisture damage. Exact measurements for temperature, moisture, and the degree of structural damage were not collected.

## Data Availability

Data including Excel sheets and Figures can be made available upon special request, although they may not be republished elsewhere. Access to the ArcGIS database is not possible because this would be an ethics violation.

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
