# Peer review of "A Citizen Science Approach to Identifying Indoor Environmental Barriers to Optimal Health for under 5s Experiencing Homelessness in Temporary Accommodation"

_ijerph, 2022, doi:10.3390/ijerph19073976_

Round 1

Reviewer 1 Report

The aims of this study are to describe the indoor environmental barriers to optimal health for U5TA and to explore the suitability of a citizen science approach. This study is interesting and offers essential information to its field of research. There are several concerns of this study.

Study aims:

The second aim is to explore the suitability of the citizen science approach to address the primary problem. However,

  1. There was no description about how to assess the suitability of this approach in a standard way and no outcomes have been mentioned in the results part about this aim.
  2. What if it comes out the citizen science approach is not suitable for this study, then how can we ensure the validity of the primary outcomes?

Thus, please reconsider whether to include “suitability of the citizen science approach” as a study aim. If yes, please provide the aforementioned information.

Title:

  1. The title was unclear. Please rewrite the title to be consistent with the study aim.
  2. “U5TA” has been mentioned throughout the content instead of “under 5s experiencing homelessness”. Please reconsider the title of whether homelessness can be fully represented by living in TA. Otherwise, a revision of title shall be done.

Introduction:

  1. Is citizen science one form of community-based participatory research (CBPR)? Please add a description about its relationship with CBPR

Methods:

  1. The sampling was conducted within one location and through only one local charity. Please add a discussion regarding the representativeness of the sample and the generalizability of the study.
  2. Whether the establishment of the WhatsApp group and allowing the participants to communicate with each other would affect the data collection, as well as the confidentiality? Please discuss.
  3. The manuscript mentioned that only three of the five collaborative meetings were audio-recorded. Please provide the reasons for the miss-recording of the other two meetings and the influence on the study results.
  4. There was no description about the analysis of the observation data. Please add a discussion about the analysis process for observational house visit data.

Results:

  1. Please discuss the possible reasons for incomplete entries for the mobile app housing survey.
  2. There was no description about data saturation. Please state whether the data saturation has been discussed among the researchers.
  3. Please discuss how to deal with the overlaps appeared among the themes and whether a combination of the themes was needed.
  4. Please discuss the influence of losing the social-demographic data.

Reviewer 2 Report

1. What is the main question addressed by the research?
This study examined the home environments of families with children under 5 who were precariously housed.

2. Do you consider the topic original or relevant in the field, and if
so, why?
Early development is important and the physical environments of impoverished children deserve greater scrutiny.

3. What does it add to the subject area compared with other published
material?
The novelty is in the citizen science approach that was used and the triangulation across methods and types of data (photos, surveys, meetings, home visits).   

4. What specific improvements could the authors consider regarding the
methodology?
This study was well done and written. The only improvements I can suggest are:
-provide some descriptive statistics to charaterise the sample demographics.
-please provide a couple more explanatory statements about the background of the HHSRS.
-p. 8: what does "+/NA" mean? 
-line 275: there's no negative sign in the table- I assume that every theme was also a hazard?
-Could the photos be better labeled and organised to clarify the purpose of their display?  

5. Are the conclusions consistent with the evidence and arguments
presented and do they address the main question posed?
Conclusions are appropriate and provide ample evidence of health hazards for these households.

6. Are the references appropriate?
Yes.

7. Please include any additional comments on the tables and figures.
n/a

Reviewer 3 Report

Thank you for the opportunity to review this manuscript. It demonstrates an interesting and novel application of citizen science methods and has elucidated important concerns of citizens living in temporary accommodation. To increase the impact of the paper, I would like to see more detailed description of the methods used and further discussion of the successes and challenges related to the use of citizen science approaches. This would be an important addition to the literature for others hoping to work with vulnerable groups.

Introduction

It would be useful for the authors to discuss why they think this study is as a “citizen science” study rather than another type of participatory research, drawing on definitions in the literature. The authors could refer to ECSA’s 10 principles of citizen science to show how the study meets these principles.

Methods

Section ‘Citizen Science Approach’ line 119. In this section, could the authors provide more information about how the workshop was run, its aims and the decisions that were made? Were participants in the workshop involved in defining the specific research questions or aims for the project or just in the design? Could the authors also refer to the wider citizen science literature to explain how decisions (e.g. about the early involvement of stakeholders) are grounded in best practice in the field?

Similarly, in the subsequent sections about the different methods used, it would be useful to understand which design decisions were informed by the participants and other stakeholders and why. For each of these methods, it would also be useful to define what the specific aims (and research questions) were and if these arose from the initial workshop.

Results

Line 392 what is meant by an alley way here? This usually means a walkway between two buildings.

Discussion

In the section ‘benefits of citizen science’:

Line 503 – how were participants involved in the analysis of data? Please clarify this in the methods section, perhaps in a fuller explanation of what was covered in the collaborative meetings.

It would be very useful to discuss some of the ethical considerations of the study in more detail. This would be useful for others seeking to conduct citizen science with vulnerable groups. For example, giving people mobile phones and then taking them away; collecting sensitive and potentially distressing information; asking people with a lot of other demands on their time to contribute to a study; payment of participants.

Line 510 – how were participants engaged in informing policy and practice? It’s not clear how the results of the study have been translated into policy and practice. Again, this could be considered an ethical consideration if nothing is done with the results of a study that vulnerable people have committed their time to or if participants are told something will happen as a result of their participation but this has not been designed into the project.

Some of the challenges related to the use of citizen science approaches used are alluded to throughout the discussion and in the conclusion. It would be useful to bring all of these together into a section of the discussion focused on the successes and challenges of the approach, including discussion of the ethical implications mentioned above. This would be useful to highlight some lessons learnt for others wanting to conduct citizen science with vulnerable groups.

Proofreading

Some proof reading needed e.g. the sentence beginning ‘Participants were recruited…’ on line 110 needs some editing. ‘Info’ instead of ‘information’ on line 182. Line 193 should read ‘Each dataset was…’. Line 212 references appear as [27], [40] instead of [27,40]. These are just some examples – a full proofread is required.

Images don’t always follow the correct order e.g. in the paragraph beginning on line 362 imahe 9 comes before image 8.

The tables could be formatted more clearly e.g. text alignment in Table 1 is not consistent; Table 2 doesn’t need the semi-colons; Table 3 + signs are not aligned and some statements have fullstops while others do not.

Round 2

Reviewer 1 Report

Authors have revised the manuscript according the reviewers' comments.